# Reversing insufficient photothermal therapy-induced tumor relapse and metastasis by regulating cancer-associated fibroblasts

Xin Li[1,6], Tuying Yong [1,2,3,4,6✉], Zhaohan Wei[1], Nana Bie[1], Xiaoqiong Zhang[1], Guiting Zhan[1], Jianye Li[1], Jiaqi Qin[1], Jingjing Yu[5], Bixiang Zhang[5], Lu Gan [1,2,3,4✉] & Xiangliang Yang [1,2,3,4✉]

Insufficient tumor accumulation and distribution of photosensitizers as well as low antitumor immunity severely restrict the therapeutic efficacy of photothermal therapy (PTT). Cancer-associated fibroblasts (CAFs) play a key role in tumor extracellular matrix (ECM) remodeling and immune evasion. Reshaping tumor microenvironment via CAF regulation might provide a potential approach for complete tumor elimination in combination with PTT. Here, tumor cell-derived microparticles co-delivering calcipotriol and Indocyanine green (Cal/ICG@MPs) are developed to modulate CAFs for improved PTT efficacy. Cal/ICG@MPs efficiently target tumor tissues and regulate CAFs to reduce tumor ECM, resulting in enhanced tumor accumulation and penetration of ICG to generate strong PTT efficacy and activate CD8[+] T cell-mediated antitumor immunity. In addition, Cal/ICG@MPs-triggered CAF regulation enhances tumor infiltration of CD8[+] T cells and ameliorates CAF-induced antigen-mediated activation-induced cell death of tumor-specific CD8[+] T cells in response to PTT, eliciting long-term antitumor immune memory to inhibit tumor recurrence and metastasis. Our results support Cal/ICG@MPs as a promising drug to improve PTT efficacy in cancer treatment.

[1] National Engineering Research Center for Nanomedicine, College of Life Science and Technology, Huazhong University of Science and Technology, Wuhan 430074, China. [2] Key Laboratory of Molecular Biophysics of the Ministry of Education, College of Life Science and Technology, Huazhong University of Science and Technology, Wuhan 430074, China. [3] Hubei Key Laboratory of Bioinorganic Chemistry and Materia Medica, Huazhong University of Science and Technology, Wuhan 430074, China. [4] Hubei Engineering Research Center for Biomaterials and Medical Protective Materials, Huazhong University of Science and Technology, Wuhan 430074, China. [5] Hepatic Surgery Center, Tongji Hospital, Tongji Medical College, Huazhong University of Science and Technology, Wuhan 430030, China. [6] These authors contributed equally: Xin Li, Tuying Yong. ✉email: yongty2018@hust.edu.cn; lugan@mail.hust.edu.cn; yangxl@mail.hust.edu.cn

Photothermal therapy (PTT), utilizing photosensitizers to generate sufficient heat under near-infrared (NIR) light irradiation to kill tumor cells and induce antitumor immunity, has emerged as a promising approach for effective cancer treatment with high selectivity and minimal invasiveness[1,2]. However, the therapeutic efficacy of PTT in solid tumors, especially large solid tumors, is compromised by the limited penetration depth of light as well as insufficient intratumoral accumulation and distribution of photosensitizers due to the complex physiological and pathological barriers in tumor tissues[3–6], generating the incomplete tumor eradication. The PTT-induced antitumor immunity is not robust enough to control the growth of the residual tumor cells in combination with tumor immunosuppressive microenvironment[7], resulting in tumor recurrence and metastasis[8,9]. Therefore, it is highly desirable to develop a combinational therapeutic strategy to improve the treatment efficacy of PTT.

Cancer-associated fibroblasts (CAFs), one of the most abundant stromal components in the tumor microenvironment of solid tumors, play critical roles in tumor occurrence, development, metastasis, therapeutic resistance, and immune evasion through the production of multiple extracellular matrix (ECM) proteins and regulatory molecules[10–12]. CAFs can build up and remodel ECM structure, forming a physical barrier that impedes the delivery of therapeutic agents and tumor infiltration of cytotoxic CD8$^+$ T cells[13–15]. In addition, CAFs secrete large quantities of growth factors and proinflammatory cytokines and chemokines, such as transforming growth factor-β (TGF-β), interleukin-6 (IL-6), and CC-chemokine ligand 2 (CCL2), to reduce T-cell response and recruit immunosuppressive cells to assist in tumor immune evasion[16–18]. Recently, it was reported that CAFs promote tumor immune escape by inducing antigen-mediated activation-induced cell death (AICD) of tumor-specific cytotoxic T lymphocytes (CTLs), in which CAFs process and cross-present tumor antigens and then kill CTLs in an antigen-specific, antigen-dependent manner via PD-L2 and FasL to protect tumor cells from immune destruction[19]. Thus, regulating CAFs might be a promising strategy to improve the therapeutic efficacy of PTT in cancer treatment. Nowadays, PTT-triggered CAF depletion by constructing CAF-targeting photothermal materials was developed to normalize tumor stiffness for enhanced cancer treatment[20,21]. However, CAF depletion may run the risk of eliminating vital stromal components that are required for tissue homeostasis, further aggravating the tumor growth[22,23]. CAF reprogramming via deactivation, rather than targeted depletion, may be the preferable therapeutic paradigm for cancer therapy[23]. Calcipotriol (Cal), a vitamin-D receptor (VDR) ligand, was found to significantly weaken the capacity of CAFs to support tumor growth via transcriptionally deactivating CAFs to a quiescent state with increased lipid droplet storage and decreased expression of fibroblast activation marker α-smooth muscle actin (α-SMA)[24,25]. However, how to achieve the targeted delivery of Cal and photosensitizers to tumor tissues to improve the therapeutic effects remains a big challenge.

Cell microparticles (MPs), extracellular vesicles with a diameter of 100–1000 nm, are shed by cells in response to various endogenous or exogenous stimulations[26–28]. Tumor cell-derived MPs hold great potential as tumor-targeted drug-delivery carriers due to excellent stability, high biocompatibility, low immunogenicity, and target-homing capability[29–31]. Here, tumor cell-derived MPs co-delivering Indocyanine green (ICG), a NIR fluorescence dye that has been approved by Food and Drug Administration (FDA) in the United State[32,33], and Cal (denoted as Cal/ICG@MPs) are specifically targeted to tumor tissues for enhanced antitumor treatment (Fig. 1). Cal/ICG@MPs efficiently regulate intratumoral CAFs and reduce tumor ECM, resulting in the enhanced tumor accumulation and penetration of ICG. Upon 808 nm laser irradiation, Cal/ICG@MPs induce strong photothermal efficacy and elicit immunogenic cell death (ICD) of tumor cells, promoting dendritic cell (DC) maturation and CD8$^+$ T-cell activation. Meanwhile, Cal/ICG@MPs-triggered CAF regulation efficiently increases intratumoral distribution of CD8$^+$ T cells into the deep tumor tissues. Cal/ICG@MPs-triggered CAF regulation ameliorates CAF-induced antigen-mediated AICD of CD8$^+$ T cells in response to PTT, facilitating T-cell proliferation and inhibiting T-cell apoptosis. Thus, intravenous injection of Cal/ICG@MPs followed by local 808 nm laser irradiation exhibits a strong systemic antitumor immune response and induces a long-term immunological memory function, generating excellent antitumor effects against not only primary large tumors, but also untreated distant tumors and even metastatic tumors via abscopal effects.

## Results

**Preparation and characterization of Cal/ICG@MPs.** To acquire Cal-, ICG-load MPs (donated as Cal@MPs or ICG@MPs, respectively) or Cal/ICG@MPs, H22 cells were irradiated with ultraviolet (UV) for 1 h, followed by treatment overnight with 4 μg mL$^{-1}$ Cal or/and 100 μg mL$^{-1}$ ICG, which was optimized according to the required drug loading efficiency (Supplementary Tables 1, 2). The obtained drug loading capacity of Cal/ICG@MPs was 3.71 ng Cal per μg protein and 0.27 μg ICG per μg protein (MPs were quantified according to the protein content) by high-performance liquid chromatography (HPLC) and UV spectrophotometer, respectively. Dynamic light scattering (DLS) analysis showed that Cal/ICG@MPs have a diameter of about 380 nm (Fig. 2a) and zeta potential of −11.4 mV (Fig. 2b), similar to blank MPs, Cal@MPs and ICG@MPs. Transmission electron microscopy (TEM) showed that Cal/ICG@MPs were monodisperse and irregularly spherical (Fig. 2c). ICG, Cal and ICG mixture (Cal/ICG), ICG@MPs, and Cal/ICG@MPs exhibited similar UV absorption spectra, confirming the successful loading of ICG into MPs (Fig. 2d). Cal/ICG@MPs exhibited a pH-responsive sustained drug release profile (Supplementary Fig. 1). No significant change in diameter (Fig. 2e and Supplementary Fig. 2a) and zeta potential (Fig. 2f and Supplementary Fig. 2b) was detected when Cal/ICG@MPs were incubated in phosphate-buffered saline (PBS) with or without 10% fetal bovine serum (FBS) for 7 days, suggesting the good stability of Cal/ICG@MPs. Cal/ICG@MPs exhibited strong temperature rise profiles in a concentration-dependent manner upon 808 nm laser irradiation (1.0 W cm$^{-2}$, Fig. 2g and Supplementary Fig. 3). Similar temperature increases induced by ICG@MPs and Cal/ICG@MPs upon 808 nm laser irradiation showed that Cal loading did not affect the photothermal efficiency of ICG@MPs (Fig. 2g and Supplementary Fig. 3).

**Cal/ICG@MPs-triggered PTT elicits ICD and activates CD8$^+$ T cell-mediated antitumor immunity.** To explore the biological function of Cal/ICG@MPs, the interaction of Cal/ICG@MPs with murine H22 hepatocarcinoma cells was first investigated. Intracellular ICG fluorescence increased in a time-dependent manner when H22 cells were treated with ICG, Cal/ICG, ICG@MPs, or Cal/ICG@MPs derived from H22 cells (Supplementary Fig. 4a). However, ICG@MPs and Cal/ICG@MPs displayed higher intracellular ICG accumulation than free ICG and Cal/ICG (Supplementary Fig. 4a), revealing that tumor cell-derived MPs enhanced cellular uptake of ICG. Consistently, stronger cytotoxicity was detected in the ICG@MPs- and Cal/ICG@MPs-treated H22 cells upon 808 nm laser irradiation compared with the free ICG- or Cal/ICG-treated group (Fig. 3a and Supplementary Fig. 4b, c). Similar results were detected in murine 4T1 breast cancer cells (Supplementary Fig. 5a–c) and human hepatocellular carcinoma HepG2 cells (Supplementary Fig. 6a, b) using Cal/ICG@MPs

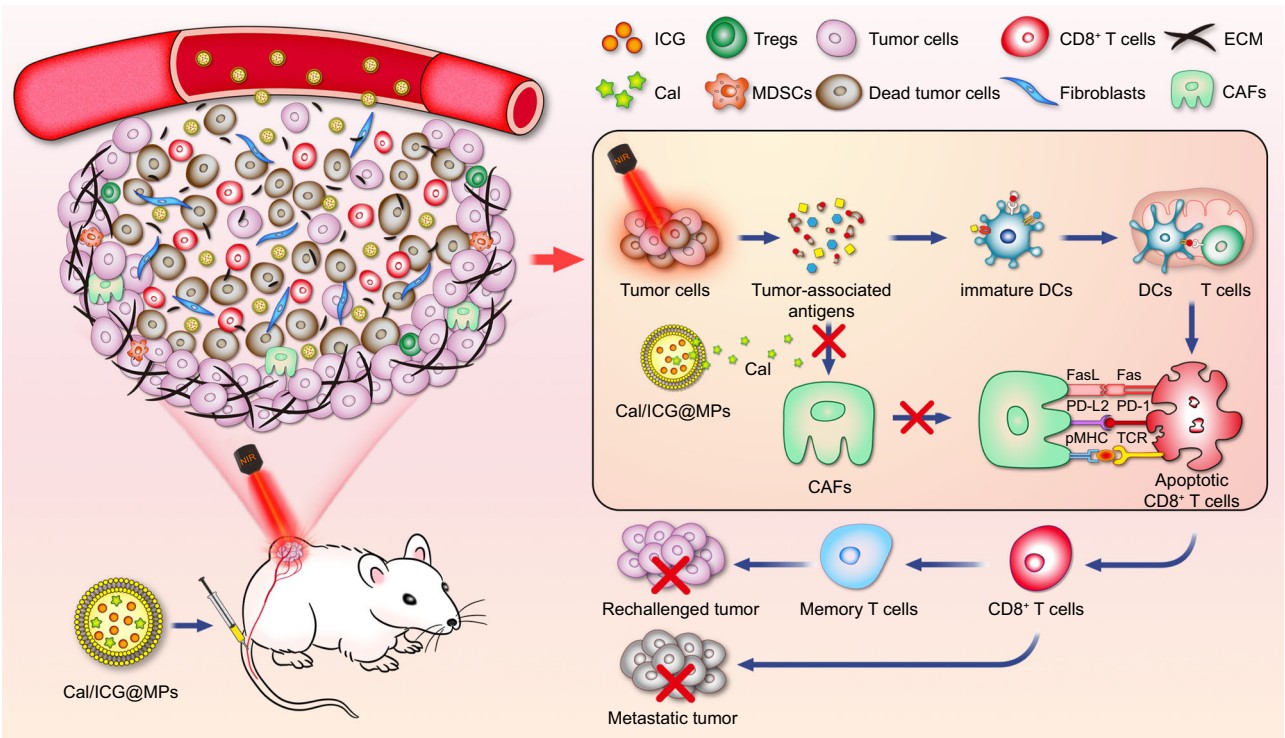

**Fig. 1 Scheme of Cal/ICG@MPs as an efficient drug to regulate CAFs to enhance PTT efficacy.** Cal/ICG@MPs efficiently target tumor tissues and regulate CAFs. (1) Cal/ICG@MPs-induced CAF regulation reduces tumor ECM to enhance the tumor accumulation and penetration of ICG, leading to the enhanced PTT effects and inducing ICD of tumor cells to activate CD8[+] T-cell-mediated antitumor immunity. (2) Cal/ICG@MPs-induced CAF regulation efficiently recruits CD8[+] T cells, promotes tumor infiltration of CD8[+] T cells, and attenuates CAF-induced antigen-mediated AICD of tumor-specific CD8[+] T cells in response to PTT. (3) Cal/ICG@MPs generate strong antitumor immunity and elicit long-term immune memory upon 808 nm laser irradiation, inhibiting tumor recurrence and metastasis after PTT.

derived from 4T1 and HepG2 cells, respectively, confirming that Cal/ICG@MPs-triggered PTT induced strong cytotoxicity against tumor cells. Cal/ICG@MPs alone did not significantly affect cell viability of H22, 4T1, and HepG2 cells (Fig. 3a and Supplementary Figs. 4b, c, 5a–c, 6b), suggesting their good biocompatibility. Cal/ICG@MPs were internalized into H22 cells in an energy-dependent manner, and clathrin-mediated endocytosis and macropinocytosis were involved in the internalization of Cal/ICG@MPs (Supplementary Fig. 7).

It is well known that PTT can trigger ICD of tumor cells to induce DC maturation and elicit antitumor immune response[34–36]. To determine whether Cal/ICG@MPs-mediated PTT could induce effective ICD effects, H22 cells were treated with PBS, free ICG, Cal/ICG, ICG@MPs, Cal@MPs, or Cal/ICG@MPs in the presence or absence of 808 nm laser irradiation, and the hallmarks of ICD including surface calreticulin (CRT) expression and release of high mobility group box 1 (HMGB1) and ATP were then detected[37–39]. As expected, the percentage of CRT-positive cells in ICG@MPs- and Cal/ICG@MPs-treated groups upon 808 nm laser irradiation were significantly higher than free ICG- and Cal/ICG-treated groups by flow cytometry (Fig. 3b) and confocal microscopy (Supplementary Fig. 8). Consistently, more secretion of HMGB1 (Fig. 3c) and ATP (Fig. 3d) was detected in ICG@MPs- and Cal/ICG@MPs-treated groups upon 808 nm laser irradiation, confirming that Cal/ICG@MPs-triggered PTT can induce strong ICD effects in H22 cells. The strong ICD induced by Cal/ICG@MPs-mediated PTT was further confirmed in 4T1 (Supplementary Fig. 5d–f) and HepG2 cells (Supplementary Fig. 6c).

DCs play a key role in the uptake of immunostimulatory damage-associated molecular patterns released by dying tumor cells and the subsequent presentation of tumor antigens to activate CD8[+] T cells[40]. To determine DC maturation induced by Cal/ICG@MPs with 808 nm laser irradiation-treated tumor cells, H22 cells were treated with PBS, free ICG, Cal/ICG, ICG@MPs, Cal@MPs, or Cal/ICG@MPs in the presence or absence of 808 nm laser irradiation, and the cell supernatants from the above treated H22 cells were then cultured with mouse bone marrow-derived DCs (BMDCs) for 24 h. Stronger expression of co-stimulatory molecules CD80 and CD86 was detected in BMDCs incubated with ICG@MPs- and Cal/ICG@MPs-treated groups upon 808 nm laser irradiation compared with other groups (Fig. 3e, f), confirming that Cal/ICG@MPs-mediated PTT efficiently triggered DC maturation. Furthermore, the above treated BMDCs were co-cultured with CD3[+] T cells isolated from spleens for 5 days, and the activation of CD8[+] T cells was evaluated by flow cytometry. Consistently, the higher percentage of CD8[+]IFN-γ[+] T (Fig. 3g), CD8[+]IL-2[+] T (Fig. 3h), CD8[+]TNF-α[+] T (Fig. 3i), and CD8[+]Granzyme B[+] T cells (Fig. 3j) was detected in ICG@MPs- and Cal/ICG@MPs-treated groups upon 808 nm laser irradiation, suggesting that DCs matured by Cal/ICG@MPs with 808 nm laser irradiation-treated H22 cells exhibited the strong capacity to activate CD8[+] T cells. DC maturation induced by Cal/ICG@MPs with 808 nm laser irradiation-treated 4T1 cell (Supplementary Fig. 9a, b) and the subsequent CD8[+] T-cell activation (Supplementary Fig. 9c–f) were further confirmed. These results indicate that Cal/ICG@MPs-mediated PTT efficiently enhances CD8[+] T cell-mediated antitumor immunity.

**Cal/ICG@MPs reduce tumor ECM to enhance tumor accumulation and penetration through regulating CAFs.** CAFs are known to build up and remodel tumor ECM by producing a variety

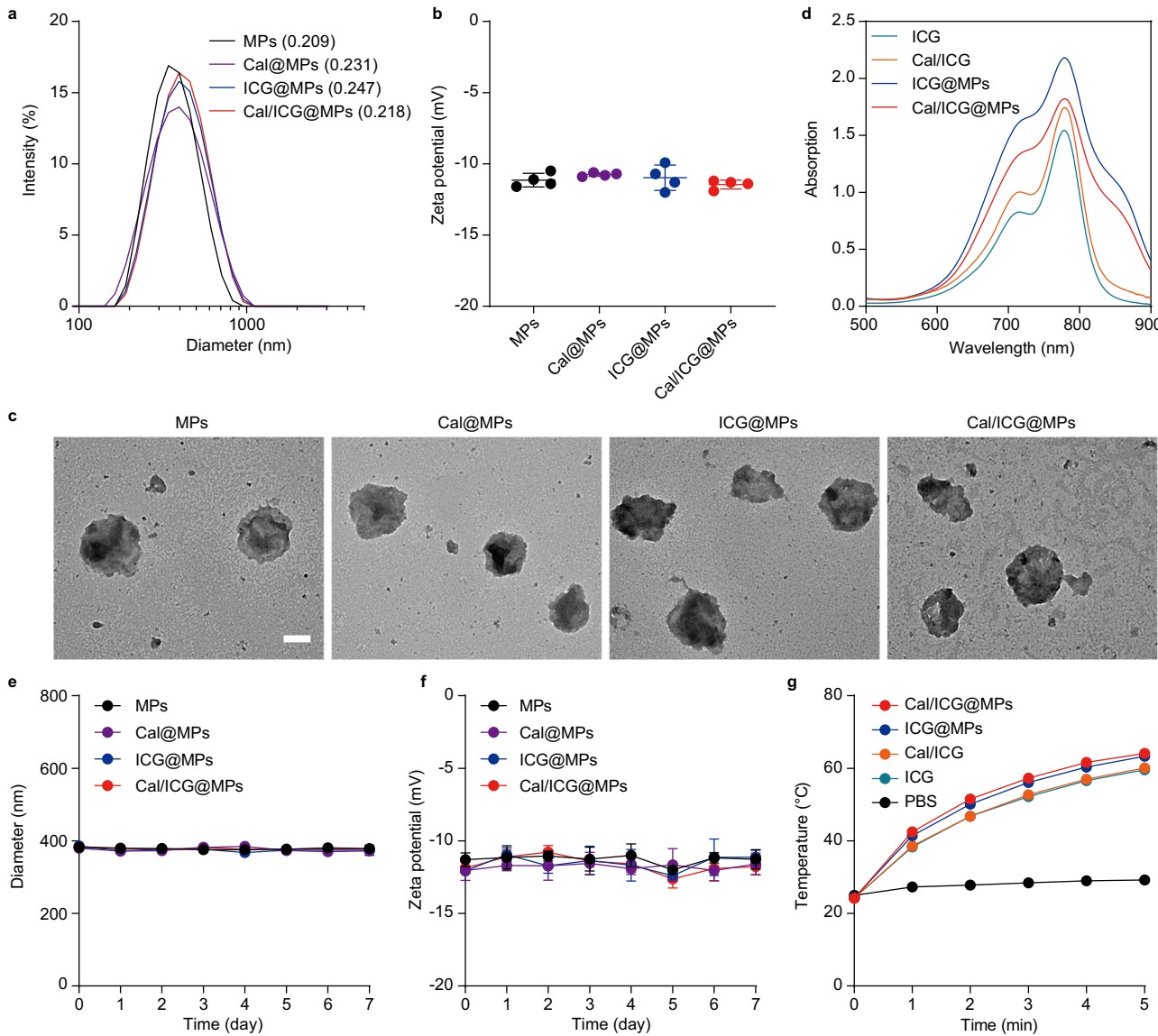

**Fig. 2 Characterization of Cal/ICG@MPs. a** Hydrodynamic diameter of MPs, Cal@MPs, ICG@MPs, and Cal/ICG@MPs by DLS analysis. Polydispersity index values are indicated in the brackets. **b** Zeta potentials of MPs, Cal@MPs, ICG@MPs, and Cal/ICG@MPs by DLS analysis. Data are presented as means ± s.d. ($n = 4$ independent samples). **c** Morphology of MPs, Cal@MPs, ICG@MPs, and Cal/ICG@MPs by TEM. Scale bar: 200 nm. **d** UV-vis spectrum spectra of ICG, Cal/ICG, ICG@MPs, and Cal/ICG@MPs. **e, f** Diameters (**e**) and zeta potentials (**f**) of MPs, Cal@MPs, ICG@MPs, and Cal/ICG@MPs after incubation in PBS containing 10% FBS for different time intervals. Data are presented as means ± s.d. ($n = 4$ independent samples). **g** Temperature curves of PBS, ICG, Cal/ICG, ICG@MPs, and Cal/ICG@MPs at the ICG concentration of 50 μg mL$^{-1}$ upon 808 nm laser irradiation (1 W cm$^{-2}$, 5 min). **a, c, d, g** show representative results of three independent samples. Source data are provided as a Source Data file.

of collagens and fibronectin[41,42]. To determine whether Cal/ICG@MPs could regulate CAFs, normal fibroblasts from suckling mouse skins were first activated with TGF-β to myofibroblasts, which expressed typical CAF markers including fibroblast activation protein alpha (FAP) and α-SMA[11,12] (Supplementary Fig. 10a–c), and then treated with ICG, Cal, Cal/ICG, ICG@MPs, Cal@MPs, or Cal/ICG@MPs. As expected, Cal and Cal/ICG significantly decreased the expression of fibronectin and α-SMA in myofibroblasts by western blot, while free ICG and ICG@MPs had no effects on the myofibroblast regulation state (Fig. 4a and Supplementary Fig. 11a, b). Cal/ICG@MPs and Cal@MPs exhibited stronger inhibition in the expression of fibronectin and α-SMA (Fig. 4a and Supplementary Fig. 11a, b), which might be due to the enhanced Cal internalization in these groups (Supplementary Fig. 12). Cal/ICG@MPs-induced decreased expression of fibronectin and α-SMA was further confirmed by immunofluorescence analysis (Supplementary Fig. 13a–c).

Collagen content determination also showed that Cal/ICG@MPs and Cal@MPs significantly decreased collagen content in myofibroblasts compared with free Cal and Cal/ICG (Fig. 4b and Supplementary Fig. 14). The similar phenomena were detected in TGF-β-activated primary human lung fibroblasts (Supplementary Fig. 15). These results revealed that Cal/ICG@MPs might efficiently remodel myofibroblasts to a quiescent state to reduce tumor ECM. Furthermore, Cal/ICG@MPs-induced CAF regulation was confirmed in organotypic tumor slices from liver cancer patients, as evidenced by the decreased expression of α-SMA, fibronectin, and collagen-I by immunofluorescence analysis and collagen fiber deposition by Masson's trichrome staining after Cal/ICG@MPs treatment (Supplementary Fig. 16a, b).

To further verify the tumor ECM reduction capacity of Cal/ICG@MPs in vivo, stroma-rich H22 tumor-bearing mice were constructed to mimic the clinical solid tumors by subcutaneous co-

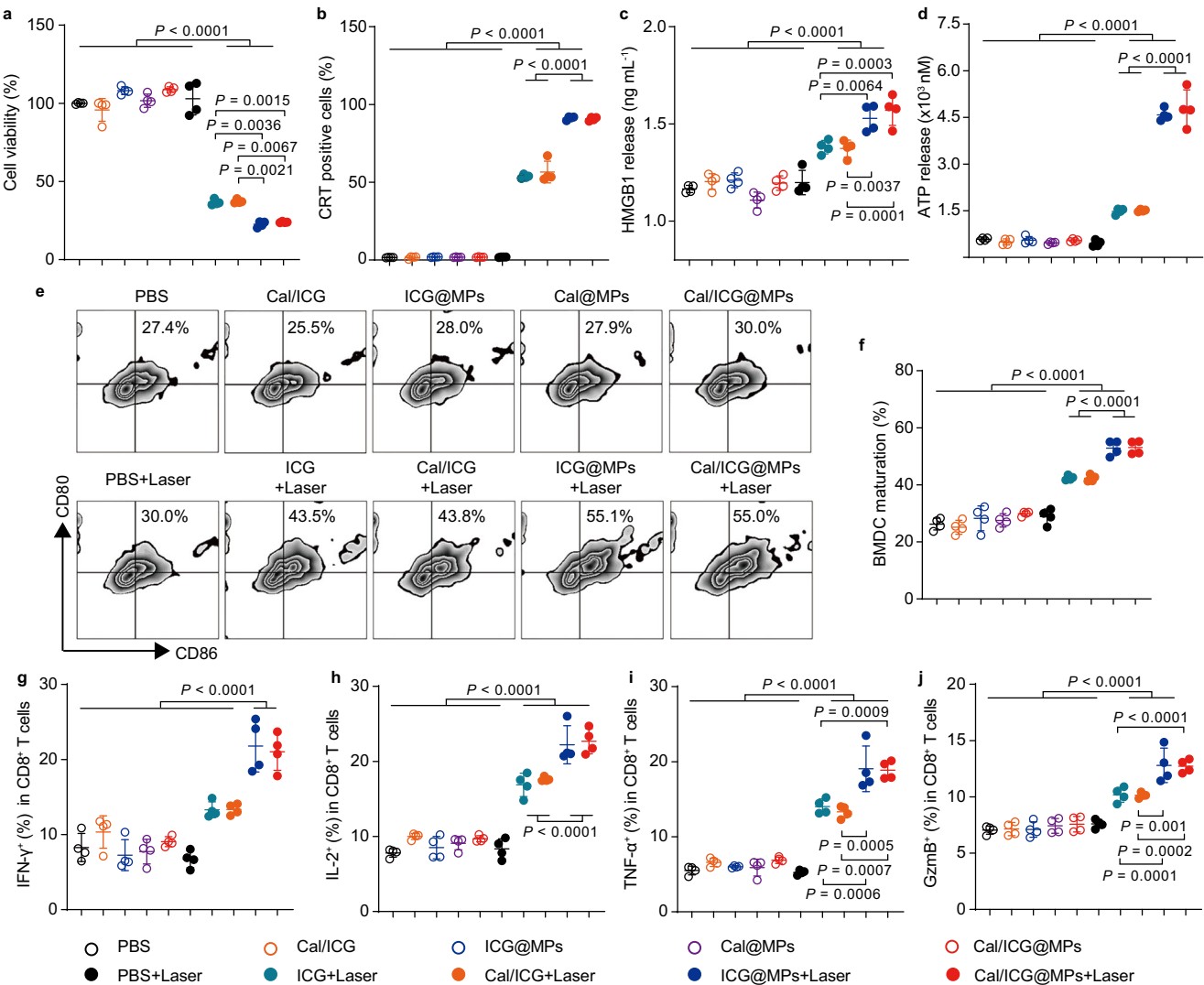

**Fig. 3 In vitro ICD effects, DC maturation, and T-cell activation triggered by Cal/ICG@MPs upon 808 nm laser irradiation. a** Cell viability of H22 cells after treatment with PBS, ICG, Cal/ICG, ICG@MPs, Cal@MPs, or Cal/ICG@MPs derived from H22 cells at the ICG concentration of 4 µg mL$^{-1}$ and Cal concentration of 60 ng mL$^{-1}$ for 4 h in the presence or absence of 808 nm laser irradiation (1 W cm$^{-2}$, 5 min). Data are presented as means ± s.d. ($n = 4$ biologically independent samples; one-way ANOVA followed by Tukey's HSD post hoc test). **b** CRT exposure on the above treated H22 cells by flow cytometry. Data are presented as means ± s.d. ($n = 4$ biologically independent samples; one-way ANOVA followed by Tukey's HSD post hoc test). **c**, **d** HMGB1 extracellular release (**c**) and ATP extracellular secretion (**d**) from the above treated H22 cells by using the HMGB1 ELISA kit and chemiluminescence ATP determination kit, respectively. Data are presented as means ± s.d. ($n = 4$ biologically independent samples; one-way ANOVA followed by Tukey's HSD post hoc test). **e**, **f** Representative flow cytometric plot (**e**) and percentages (**f**) of matured BMDCs after the immature BMDCs were co-cultured with the cell supernatants from the above treated H22 cells for 24 h. Data are presented as means ± s.d. ($n = 4$ biologically independent samples; one-way ANOVA followed by Tukey's HSD post hoc test). **g–j** Percentages of IFN-γ$^+$ (**g**), IL-2$^+$ (**h**), TNF-α$^+$ (**i**), and Granzyme B (GzmB)$^+$ (**j**) cells in CD8$^+$ T cells after CD3$^+$ T cells were incubated with the above-matured BMDCs for 5 days by flow cytometry. Data are presented as means ± s.d. ($n = 4$ biologically independent samples; one-way ANOVA followed by Tukey's HSD post hoc test). Source data are provided as a Source Data file.

injection of TGF-β-activated skin fibroblasts and H22 cells at a ratio of 1:2, as evidenced by high expression of α-SMA, fibronectin and collagen-I in tumor tissues (Supplementary Fig. 17a) with faster tumor formation (Supplementary Fig. 17b-d). These mice were intravenously injected with PBS, ICG, Cal/ICG, ICG@MPs, Cal@MPs, or Cal/ICG@MPs twice every 2 days, followed with or without 808 nm laser irradiation. Consistently, flow cytometric analysis showed that Cal@MPs and Cal/ICG@MPs significantly decreased the numbers of CAFs, namely podoplanin (PDPN)$^+$CD140α$^+$CD45$^-$EpCAM$^-$CD31$^-$cells (Fig. 4c and Supplementary Fig. 18a) or PDPN$^+$α-SMA$^+$CD45$^-$EpCAM$^-$CD31$^-$cells

(Supplementary Fig. 18a, b), in tumor tissues compared with PBS group although the numbers of PDPN$^+$α-SMA$^-$CD45$^-$EpCAM$^-$CD31$^-$cells also decreased (Supplementary Fig. 18a, c). Meanwhile, Cal@MPs and Cal/ICG@MPs significantly inhibited the ratios of Ki67-positive proliferative CAFs (Supplementary Fig. 18d). However, Cal@MPs and Cal/ICG@MPs did not significantly induce apoptosis of myofibroblasts (Supplementary Fig. 19), revealing that Cal/ICG@MPs might regulate CAFs by deactivation, but not depletion. In addition, 808 nm laser irradiation did not markedly promote Cal/ICG@MPs-induced decrease in the numbers of CAFs (Fig. 4c and Supplementary Fig. 18b), and myofibroblasts were resistant to Cal/

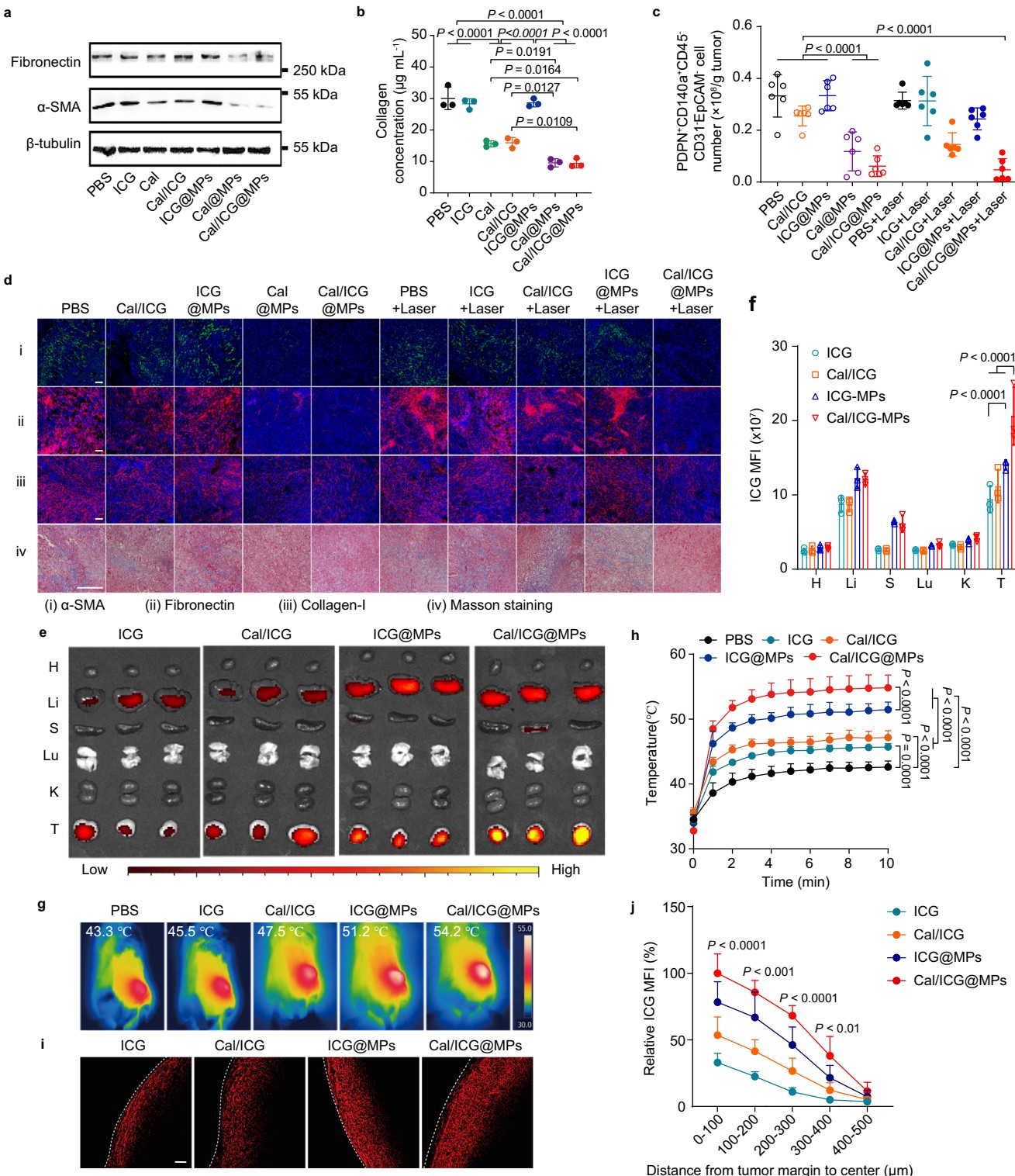

(i) α-SMA    (ii) Fibronectin    (iii) Collagen-I    (iv) Masson staining

ICG@MPs-triggered PTT (Supplementary Fig. 20), suggesting that Cal/ICG@MPs-triggered PTT did not deplete CAFs. Meanwhile, significantly lower expression of α-SMA, fibronectin, and collagen-I and decreased deposition of collagen fibers were detected in the tumors of Cal@MPs- or Cal/ICG@MPs-treated group with or without 808 nm laser irradiation (Fig. 4d and Supplementary Fig. 21). Cal/ICG@MPs-induced decrease in α-SMA expression and collagen fiber deposition was also displayed in azoxymethane (AOM)/dextran sodium sulfate (DSS)-induced colitis-associated cancer (CAC) mouse model (Supplementary Fig. 22a–c). Moreover, the solid stress of

tumor tissues in stroma-rich H22 tumor-bearing mice (Supplementary Fig. 23a) and 4T1 tumor-bearing mice (Supplementary Fig. 23b) was significantly reduced after treatment with Cal@MPs and Cal/ICG@MPs. Taken together, these data suggest that Cal/ICG@MPs efficiently reduce tumor ECM through regulating CAFs.

The dense tumor ECM severely restricts tumor accumulation and penetration of antitumor drugs, limiting their therapeutic effects[6,43]. In view of tumor ECM reduction induced by Cal/ICG@MPs-triggered CAF regulation, the tumor accumulation of Cal/ICG@MPs was first assessed in stroma-rich H22 tumor-

**Fig. 4 Cal/ICG@MPs-induced CAF regulation, ECM reduction, and enhanced tumor accumulation and penetration. a** Western blot analysis of fibronectin and α-SMA expression in myofibroblasts after treatment with PBS, ICG, Cal, Cal/ICG, ICG@MPs, Cal@MPs or Cal/ICG@MPs derived from H22 cells at the ICG concentration of 4 μg mL$^{-1}$ and Cal concentration of 60 ng mL$^{-1}$ for 48 h. Images are representative of three independent samples. **b** Relative collagen content in myofibroblasts after treatment indicated in (**a**) measured by Sirius red total collagen detection kit. Data are presented as means ± s.d. ($n = 3$ biologically independent samples; one-way ANOVA followed by Tukey's HSD post hoc test). **c** Numbers of CAFs in tumor tissues of stroma-rich H22 tumor-bearing mice at 14 days after intravenous injection of PBS, ICG, Cal/ICG, ICG@MPs, Cal@MPs, or Cal/ICG@MPs at the ICG dosage of 8 mg kg$^{-1}$ and Cal dosage of 120 μg kg$^{-1}$ twice every 2 days, followed with or without 808 nm laser irradiation (1.5 W cm$^{-2}$, 10 min) at 2 h after the last injection. Data are presented as means ± s.d. ($n = 6$ mice per group; one-way ANOVA followed by Tukey's HSD post hoc test). **d** Immunofluorescence staining of α-SMA, fibronectin, and collagen-I, and Masson's trichrome staining of collagen in tumor sections of stroma-rich H22 tumor-bearing mice after treatment indicated in **c**. Images are representative of three biologically independent mice. Scale bars: 100 μm. **e, f** Ex vivo NIRF imaging (**e**) and ICG mean fluorescence intensity (MFI, **f**) in tumors and major organs of stroma-rich H22 tumor-bearing mice at 72 h after intravenous injection of ICG, Cal/ICG, ICG@MPs, or Cal/ICG@MPs at the ICG dosage of 5 mg kg$^{-1}$ and Cal dosage of 75 μg kg$^{-1}$ twice every 2 days. Data are presented as means ± s.d. ($n = 3$ mice per group; two-way ANOVA followed by Tukey's multiple comparisons post-test). **g, h** Representative IR thermal images (**g**) and temperature curves (**h**) of stroma-rich H22 tumor-bearing mice after intravenous injection of PBS, ICG, Cal/ICG, ICG@MPs, or Cal/ICG@MPs at the ICG dosage of 8 mg kg$^{-1}$ and Cal dosage of 120 μg kg$^{-1}$ twice every 2 days, followed with 808 nm laser irradiation (1.5 W cm$^{-2}$, 10 min) at 2 h after the last injection. Data are presented as means ± s.d. ($n = 6$ mice per group; two-way ANOVA followed by Tukey's multiple comparisons post-test). **i** ICG fluorescence images in stroma-rich H22 tumors after treatment with ICG, Cal/ICG, ICG@MPs, or Cal/ICG@MPs at the ICG concentration of 4 μg mL$^{-1}$ and Cal concentration of 60 ng mL$^{-1}$ for 48 h. The tumor margin was marked with a white dotted line. Images are representative of three biologically independent mice. Scale bar: 100 μm. **j** ICG distribution profile from the margin to the center of H22 tumors after treatment indicated in **i** by Image J software. Data are presented as means ± s.d. ($n = 9$ fields in total from three mice; two-way ANOVA followed by Tukey's multiple comparisons post test). Source data and exact $P$ values for **j** are provided as a Source Data file.

bearing mice constructed by co-injection of H22 cells and TGF-β-activated skin fibroblasts. The mice were intravenously injected with ICG, Cal/ICG, ICG@MPs, or Cal/ICG@MPs, and ICG fluorescence was then detected using in vivo imaging system (Supplementary Fig. 24). ICG@MPs- and Cal/ICG@MPs-treated groups exhibited stronger ICG fluorescence in tumor tissues compared with ICG- and Cal/ICG-treated groups, suggesting that tumor cell-derived MPs efficiently enhanced tumor accumulation of ICG. However, the strongest ICG fluorescence was detected in Cal/ICG@MPs-treated group, which might be due to the Cal/ICG@MPs-induced decreased tumor ECM. To directly evaluate the enhanced tumor accumulation of Cal/ICG@MPs, the ex vivo ICG fluorescence was determined in tumors of stroma-rich H22 tumor-bearing mice at 72 h after intravenous administration (Fig. 4e, f). Consistently, Cal/ICG@MPs exhibited the highest tumor accumulation, about 1.5-, 1.8-, and 2.2-fold relative to ICG@MPs-, Cal/ICG- and free ICG-treated groups, respectively. Meanwhile, when organotypic ex vivo liver cancer patient-derived tumor slices were treated with ICG, Cal/ICG, ICG@MPs or Cal/ICG@MPs derived from HepG2 cells, the strongest ICG fluorescence was also detected in Cal/ICG@MPs-treated group (Supplementary 25a, b). The enhanced tumor accumulation of Cal/ICG@MPs was further confirmed by the higher in vivo photothermal efficacy in stroma-rich H22 tumor-bearing mice constructed by co-injection of H22 cells and TGF-β-activated skin fibroblasts (Fig. 4g, h) or hepatic stellate cells (HSCs, Supplementary Fig. 26a, b), as well as stroma-rich 4T1 tumor-bearing mice constructed by co-injection of 4T1 cells and TGF-β-activated skin fibroblasts (Supplementary Fig. 27a, b), as evidenced by a maximal temperature in tumor tissues of Cal/ICG@MPs-treated group upon 808 nm laser irradiation for 10 min. Cal/ICG@MPs was found to be efficiently accumulated in tumor cells and CAFs. However, more Cal/ICG@MPs were accumulated in tumor cells compared with those in CAFs (Supplementary Fig. 28a, b). Furthermore, the tumor penetration capacity of Cal/ICG@MPs was determined by incubating ex vivo tumor tissues of stroma-rich H22 tumor-bearing mice (Fig. 4i, j) or organotypic liver cancer patient-derived tumor slices (Supplementary Fig. 25a, c) in cell culture media containing ICG, Cal/ICG, ICG@MPs or Cal/ICG@MPs. At 48 h after incubation, the tumors were sectioned and the distribution of ICG signals from tumor margin to the interior was imaged. As expected, free ICG

was mostly distributed around the tumor periphery. More Cal/ICG@MPs were spread inside the tumor tissues compared with other groups, suggesting that Cal/ICG@MPs-induced tumor ECM remodeling might contribute to their enhanced tumor penetration.

**Cal/ICG@MPs attenuate CAF-induced antigen-mediated AICD of CD8$^+$ T cells in response to PTT.** The successful CD8$^+$ T cell-mediated antitumor immunity relies in part on the recruitment of CD8$^+$ T cells to the tumor tissues[44]. Consistent with Cal/ICG@MPs-induced stronger CAF regulation, transwell analysis showed that Cal/ICG@MPs-treated myofibroblasts exhibited a stronger ability to recruit CD8$^+$ T cells compared with free Cal- and Cal/ICG-treated groups (Supplementary Fig. 29a). However, Cal/ICG@MPs-treated myofibroblasts did not markedly affect the ratios of CD69$^+$ and IFN-γ$^+$ in CD8$^+$ T cells (Supplementary Fig. 29b, c), suggesting that Cal/ICG@MPs-induced CAF modulation contributes to CD8$^+$ T-cell recruitment, but not CD8$^+$ T-cell activation. Recent work showed that CAFs could induce antigen-mediated AICD of tumor-specific CTLs to promote tumor immune evasion by processing and cross-presenting tumor antigens via the class I major histocompatibility complex (MHC-I)[19]. To determine whether Cal/ICG@MPs-induced CAF modulation could ameliorate antigen-mediated AICD of CTLs during PTT, myofibroblasts were pre-treated with free ICG, Cal/ICG, Cal@MPs, ICG@MPs, or Cal/ICG@MPs, and then incubated with the culture supernatants of H22 cells pretreated with the corresponding same formulations in the presence or absence of 808 nm laser irradiation, respectively, followed by MHC-I expression analysis in myofibroblasts by flow cytometry (Fig. 5a). As expected, MHC-I expression in ICG- or ICG@MPs-treated group upon 808 nm laser irradiation significantly increased compared with other groups (Fig. 5b). The highest MHC-I expression was detected in ICG@MPs-treated group upon 808 nm laser irradiation, revealing that ICG@MPs-treated myofibroblasts exhibited stronger antigen cross-presentation after exposure to more tumor antigen released by ICG@MPs with 808 nm laser irradiation-treated H22 cells. Although ICG@MPs and Cal/ICG@MPs induced similar ICD effects in H22 cells upon 808 nm laser irradiation (Fig. 3b–d), MHC-I expression in Cal/ICG@MPs with 808 nm laser irradiation-treated group was significantly lower than ICG@MPs

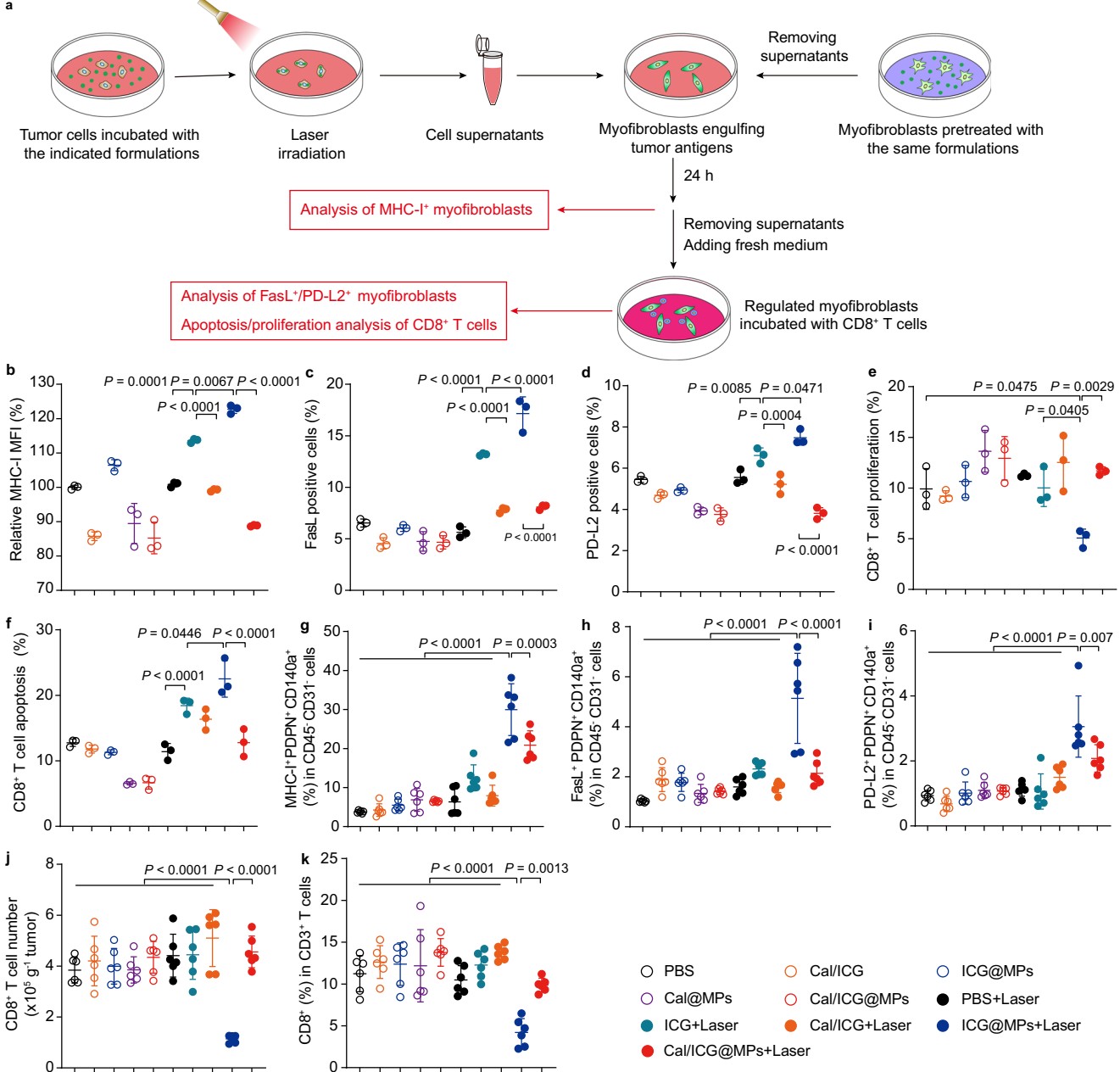

**Fig. 5 Attenuated CAF-induced antigen-mediated AICD effects of CD8$^+$ T cells by Cal/ICG@MPs. a** Scheme of attenuated myofibroblast-induced antigen-mediated AICD effect of CD8$^+$ T cells by Cal/ICG@MPs. The cell supernatants were collected after tumor cells were treated with PBS, ICG, Cal/ICG, ICG@MPs, Cal@MPs, or Cal/ICG@MPs at the ICG concentration of 4 μg mL$^{-1}$ and Cal concentration of 60 ng mL$^{-1}$ for 4 h in the presence or absence of 808 nm laser irradiation (1 W cm$^{-2}$, 5 min). Myofibroblasts were pretreated with PBS, ICG, Cal/ICG, ICG@MPs, Cal@MPs, or Cal/ICG@MPs at the ICG concentration of 4 μg mL$^{-1}$ and Cal concentration of 60 ng mL$^{-1}$ for 48 h, and then co-cultured with the above tumor cell supernatants for 24 h. Finally, the treated myofibroblasts were co-cultured with CD8$^+$ T cells for another 48 h. The phenotype of myofibroblasts and CD8$^+$ T cells was analyzed by flow cytometry. **b** MHC-I MFI in myofibroblasts after the pretreated myofibroblasts were co-cultured with the cell supernatants from the treated H22 cells indicated in **a**. Data are presented as means ± s.d. ($n$ = 3 biologically independent samples; one-way ANOVA followed by Tukey's HSD post hoc test). **c, d** FasL (**c**) and PD-L2 (**d**) expression in myofibroblasts after the pretreated myofibroblasts were co-cultured with the cell supernatants from the treated H22 cells, and then CD8$^+$ T cells indicated in **a**. Data are presented as means ± s.d. ($n$ = 3 biologically independent samples; one-way ANOVA followed by Tukey's HSD post hoc test). **e, f** Proliferation (**e**) and apoptosis (**f**) ratios of CD8$^+$ T cells after treatment are indicated in **a**. Data are presented as means ± s.d. ($n$ = 3 biologically independent samples; one-way ANOVA followed by Tukey's HSD post hoc test). **g–i** MHC-I (**g**), FasL (**h**), and PD-L2 (**i**) expression in CAFs of stroma-rich H22 tumor-bearing mice at 14 days after intravenous injection of PBS, ICG, Cal/ICG, ICG@MPs, Cal@MPs, or Cal/ICG@MPs at the ICG dosage of 8 mg kg$^{-1}$ and Cal dosage of 120 μg kg$^{-1}$ twice every 2 days, followed with or without 808 nm laser irradiation (1.5 W cm$^{-2}$, 10 min) at 2 h after the last injection by flow cytometry. Data are presented as means ± s.d. ($n$ = 6 mice per group; one-way ANOVA followed by Tukey's HSD post hoc test). **j, k** The numbers (**j**) and ratios (**k**) of CD8$^+$ T cells in tumor tissues of stroma-rich H22 tumor-bearing mice after treatment are indicated in **g**. Data are presented as means ± s.d. ($n$ = 6 mice per group; one-way ANOVA followed by Tukey's HSD post hoc test). Source data are provided as a Source Data file.

with 808 nm laser irradiation-treated group (Fig. 5b), which might be due to the fact that Cal/ICG@MPs-induced CAF regulation decreased antigen cross-presentation. In view of CAF-induced antigen-mediated AICD of CTLs by Fas/FasL signaling-mediated CTL apoptosis and PD-L2/PD-1 signaling-mediated CTL anergy[19], the above-treated myofibroblasts were incubated with CD8$^+$ T cells and the expression of FasL and PD-L2 in myofibroblasts was then detected by flow cytometry (Fig. 5a). Consistently, stronger expression of FasL and PD-L2 was detected in ICG@MPs-treated group upon 808 nm laser irradiation, and loading Cal significantly decreased ICG@MPs with 808 nm laser irradiation-induced increase in FasL and PD-L2 expression (Fig. 5c, d). Correspondingly, the higher cell proliferation rate (Fig. 5e) and lower apoptotic rate of CD8$^+$ T cells (Fig. 5f) were detected in Cal/ICG@MPs with 808 nm laser irradiation-treated group compared with ICG@MPs with 808 nm laser irradiation-treated group. Similar results were detected in Cal/ICG@MPs-pretreated myofibroblasts co-cultured with Cal/ICG@MPs with 808 nm laser irradiation-treated 4T1 cell supernatants and CD8$^+$ T cells (Supplementary Fig. 30a–c). Moreover, the expression of MHC-I (Fig. 5g), FasL (Fig. 5h), and PD-L2 (Fig. 5i) in CAFs was significantly reduced in stroma-rich H22 tumor-bearing mice after intravenous injection of Cal/ICG@MPs upon 808 nm laser irradiation compared with ICG@MPs with 808 nm laser irradiation-treated group. Meanwhile, the numbers and ratios of CD8$^+$ T cells in tumor tissues of mice undergoing ICG@MPs-triggered PTT markedly decreased, while loading Cal significantly abrogated the decrease in the numbers and ratios of CD8$^+$ T cells induced by ICG@MPs with 808 nm laser irradiation (Fig. 5j, k), further confirming that Cal/ICG@MPs-triggered CAF regulation efficiently ameliorates CAF-induced antigen-mediated AICD of CD8$^+$ T cells in response to PTT.

**Cal/ICG@MPs improve tumor immune microenvironment upon 808 nm laser irradiation**. As Cal/ICG@MPs-triggered PTT efficiently induces ICD and activates CD8$^+$ T cells, and Cal/ICG@MPs regulate CAFs to recruit CD8$^+$ T cells and decrease the antigen-mediated AICD of CD8$^+$ T cells, the tumor immune microenvironment was then investigated in stroma-rich H22 tumor-bearing mice after intravenous injection of PBS, ICG, Cal/ICG, ICG@MPs, Cal@MPs, or Cal/ICG@MPs derived from H22 cells twice every 2 days, followed with or without 808 nm laser irradiation at tumor sites for 10 min (Fig. 6a). Consistently, ICG@MPs and Cal/ICG@MPs significantly promoted DC maturation upon 808 nm laser irradiation (Fig. 6b and Supplementary 31a, 32a). The strongest effects of promoting DC maturation were detected in Cal/ICG@MPs with 808 nm laser irradiation-treated group (Fig. 6b and Supplementary 31a, 32a). Correspondingly, the highest numbers and ratios of activated CD69$^+$CD8$^+$ T (Fig. 6c and Supplementary Figs. 31a, 32b), CD8$^+$IFN-γ$^+$ T (Fig. 6d and Supplementary Figs. 31a, 32c), and CD8$^+$IL-2$^+$ T cells (Fig. 6e and Supplementary Figs. 31a, 32d) were detected in tumor tissues of Cal/ICG@MPs with 808 nm laser irradiation-treated group compared with other groups, revealing that Cal/ICG@MPs efficiently activated antitumor immunity upon 808 nm laser irradiation. Here, although the ratios of activated CD8$^+$CD69$^+$ T, CD8$^+$IFN-γ$^+$ T, and CD8$^+$IL-2$^+$ T cells in ICG@MPs with 808 nm laser irradiation-treated group were significantly increased compared with PBS group (Supplementary Fig. 32b–d), their numbers decreased while loading Cal abrogated the ICG@MPs with 808 nm laser irradiation-induced decrease in the numbers of activated CD8$^+$CD69$^+$ T, CD8$^+$IFN-γ$^+$ T and CD8$^+$IL-2$^+$ T cells (Fig. 6c–e), further confirming that CAF regulation by Cal can efficiently abolish CAF-induced antigen-mediated AICD of CD8$^+$ T cells upon ICG@MPs-triggered PTT. Cal/ICG@MPs markedly increased the ratios of CD8$^+$PD-1$^+$ T cells in

tumor tissues upon 808 nm laser irradiation (Supplementary Fig. 32e), which might be due to the PTT-generated tumor antigen stimulation[45]. Furthermore, Cal/ICG@MPs significantly decreased the ratios and numbers of regulatory T cells (Tregs, Supplementary Fig. 31a and Supplementary Fig. 32f, g), polymorphonuclear myeloid-derived suppressor cells (PMN-MDSCs, Supplementary Fig. 31a, and Supplementary Fig. 32h, i) and M2-like tumor-associated macrophages (TAMs, Supplementary Fig. 31a and Supplementary Fig. 32j, k) upon 808 nm laser irradiation, suggesting the improved tumor immunosuppressive microenvironment. Similar phenomena were detected in the tumor-draining lymph nodes (LNs, Supplementary Fig. 33) and spleens (Supplementary Fig. 34) of Cal/ICG@MPs-treated group upon 808 nm laser irradiation. Meanwhile, Cal/ICG@MPs significantly increased the levels of proinflammatory cytokines IFN-γ (Supplementary Fig. 35a) and TNF-α (Supplementary Fig. 35b) in serum upon 808 nm laser irradiation. These results indicate that Cal/ICG@MPs efficiently activate antitumor immunity and alleviate tumor immune suppressive microenvironment upon 808 nm laser irradiation.

CAF regulation not only recruits CD8$^+$T cells and influences antigen-mediated AICD of CD8$^+$ T cells, it also remodels tumor ECM to improve the intratumoral distribution of CD8$^+$ T cells[15]. To further determine the effects of Cal/ICG@MPs-induced CAF regulation on the intratumoral CD8$^+$ T cells, stroma-rich H22 tumor-bearing mice were intravenously injected with PBS, ICG, Cal/ICG, ICG@MPs, Cal@MPs or Cal/ICG@MPs twice every 2 days, followed with 808 nm laser irradiation at tumor sites for 10 min, and then the distribution of α-SMA$^+$ CAFs and CD8$^+$ T cells in tumor tissues was observed by immunofluorescence staining (Fig. 6f). Consistently, α-SMA expression was significantly decreased in Cal/ICG@MPs-treated group (Fig. 6f and Supplementary Fig. 36), confirming that Cal/ICG@MPs efficiently regulated CAFs. As expected, more CD8$^+$ T cells were detected in Cal/ICG@MPs with 808 nm laser irradiation-treated group compared with ICG@MPs with 808 nm laser irradiation-treated group (Fig. 6g). Meanwhile, CD8$^+$ T-cell numbers around α-SMA$^+$ CAFs in Cal/ICG@MPs with 808 nm laser irradiation-treated group was about fourfold as compared to ICG@MPs with 808 nm laser irradiation-treated group (Fig. 6h), further confirming that Cal/ICG@MPs attenuate CAF-induced antigen-mediated AICD of CD8$^+$ T cells upon PTT. Importantly, more CD8$^+$ T cells in Cal/ICG@MPs with 808 nm laser irradiation-treated group were distributed in deep tumor tissues compared with other groups (Fig. 6i), which might be due to the tumor ECM reduction by CAF regulation. These results suggest that Cal/ICG@MPs efficiently improve the intratumoral distribution of CD8$^+$ T cells in response to PTT.

**Cal/ICG@MPs inhibit proliferation and recurrence of large H22 tumors upon 808 nm laser irradiation**. In view of the enhanced tumor accumulation and penetration of ICG, and the improved tumor immune microenvironment, the antitumor activity of Cal/ICG@MPs against large tumors was determined in stroma-rich H22 tumor-bearing mice constructed by co-injection of TGF-β-activated skin fibroblasts and H22 cells upon 808 nm laser irradiation. When the tumor reached about 300 mm³, the mice were intravenously injected with PBS, ICG, Cal/ICG, ICG@MPs, Cal@MPs or Cal/ICG@MPs derived from H22 cells twice every 2 days, followed with or without 808 nm laser irradiation at the tumor sites for 10 min (Fig. 7a). Although ICG@MPs significantly inhibited tumor growth upon 808 nm laser irradiation at the initial stage, the tumors grew rapidly after 26 d (Fig. 7b). Cal/ICG@MPs with 808 nm laser irradiation exhibited the strongest tumor inhibition activity, with 37.5% of mice becoming tumor-free (Fig. 7b). The lowest tumor weight was also detected in Cal/

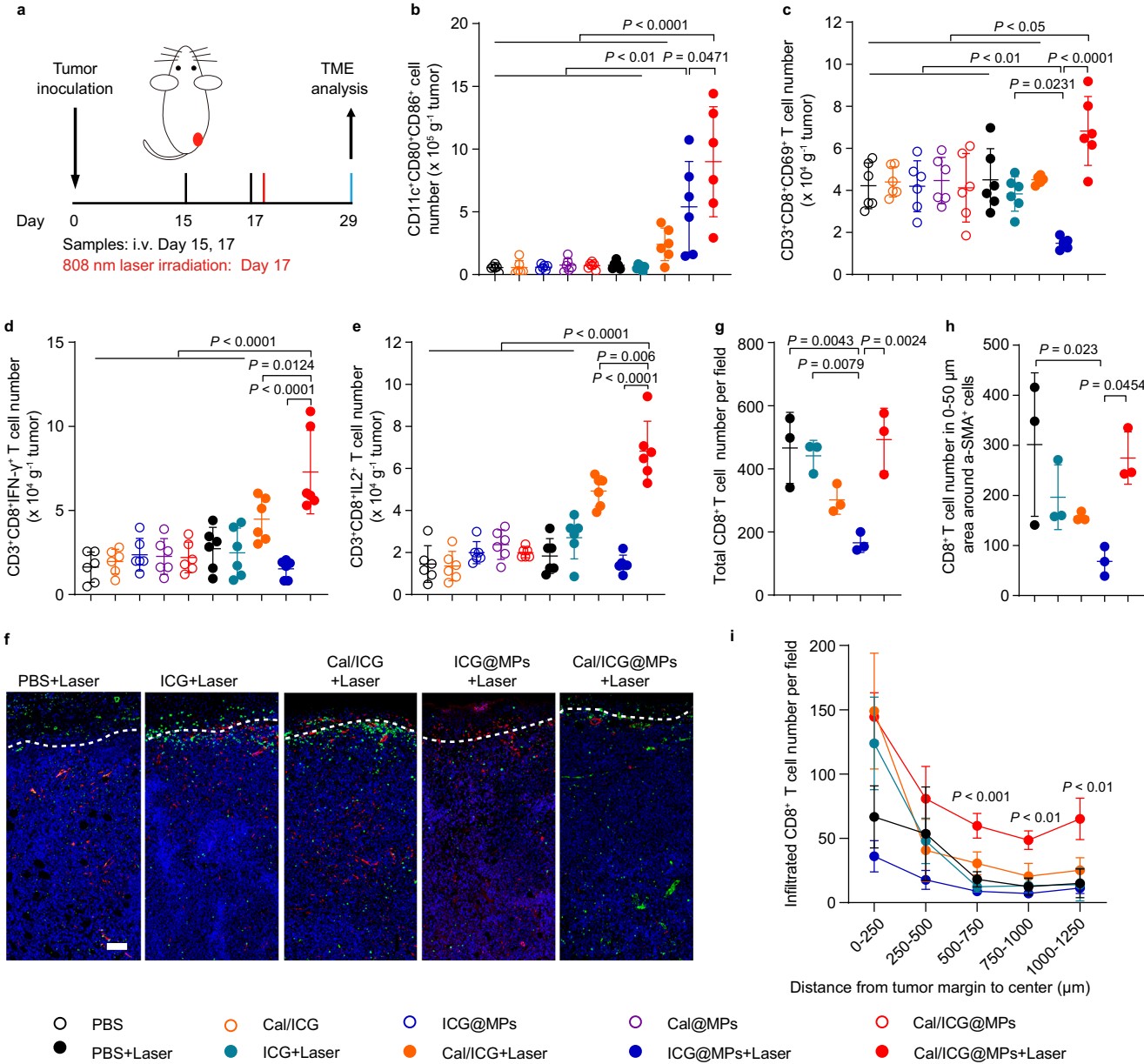

**Fig. 6 Cal/ICG@MPs-induced CD8$^+$ T-cell tumor infiltration in H22 tumor-bearing mice upon 808 nm laser irradiation. a** Schematic schedule for CD8$^+$ T-cell tumor infiltration analysis in stroma-rich H22 tumor-bearing mice after intravenous injection of PBS, ICG, Cal/ICG, ICG@MPs, Cal@MPs, or Cal/ICG@MPs derived from H22 cells at the ICG dosage of 8 mg kg$^{-1}$ and Cal dosage of 120 μg kg$^{-1}$, followed with or without 808 nm laser irradiation (1.5 W cm$^{-2}$, 10 min) at 2 h after the last injection. **b–e** Numbers of CD11c$^+$CD80$^+$CD86$^+$ cells (**b**), CD8$^+$CD69$^+$ T cells (**c**), CD8$^+$IFN-γ$^+$ T cells (**d**), and CD8$^+$IL-2$^+$ T cells (**e**) in tumor tissues of stroma-rich H22 tumor-bearing mice after treatment indicated in **a**. Data are presented as means ± s.d. ($n = 6$ mice per group; one-way ANOVA followed by Tukey's HSD post hoc test). **f** Immunofluorescence images of CD8$^+$ T cells (labeling with Cy3-conjugated CD8 antibody, green) and CAFs (labeling with Cy5-conjugated α-SMA antibody, red) in tumor tissues of stroma-rich H22 tumor-bearing mice after intravenous injection of PBS, ICG, Cal/ICG, ICG@MPs or Cal/ICG@MPs at the ICG dosage of 8 mg kg$^{-1}$ and Cal dosage of 120 μg kg$^{-1}$, followed with 808 nm laser irradiation indicated in **a**. The tumor margin was marked with a white dotted line. Images are representative of three biologically independent mice. Scale bar: 100 μm. **g** CD8$^+$ T-cell numbers in tumor tissues of stroma-rich H22 tumor-bearing mice after treatment indicated in **f** analyzed by StrataQuest software. Data are presented as means ± s.d. ($n = 3$ fields in total from two mice; one-way ANOVA followed by Tukey's HSD post hoc test). **h** CD8$^+$ T-cell numbers within 50 μm distance around CAFs in tumor tissues of stroma-rich H22 tumor-bearing mice after treatment indicated in **f**. Data are presented as means ± s.d. ($n = 3$ fields in total from two mice; one-way ANOVA followed by Tukey's HSD post hoc test). **i** CD8$^+$ T-cell numbers from the margin to center areas in tumor tissues of stroma-rich H22 tumor-bearing mice after treatment indicated in **f**. Data are presented as means ± s.d. ($n = 6$ fields in total from three mice; two-way ANOVA followed by Tukey's multiple comparisons post test). Source data and exact $P$ values for **b**, **c**, **i** are provided as a Source Data file.

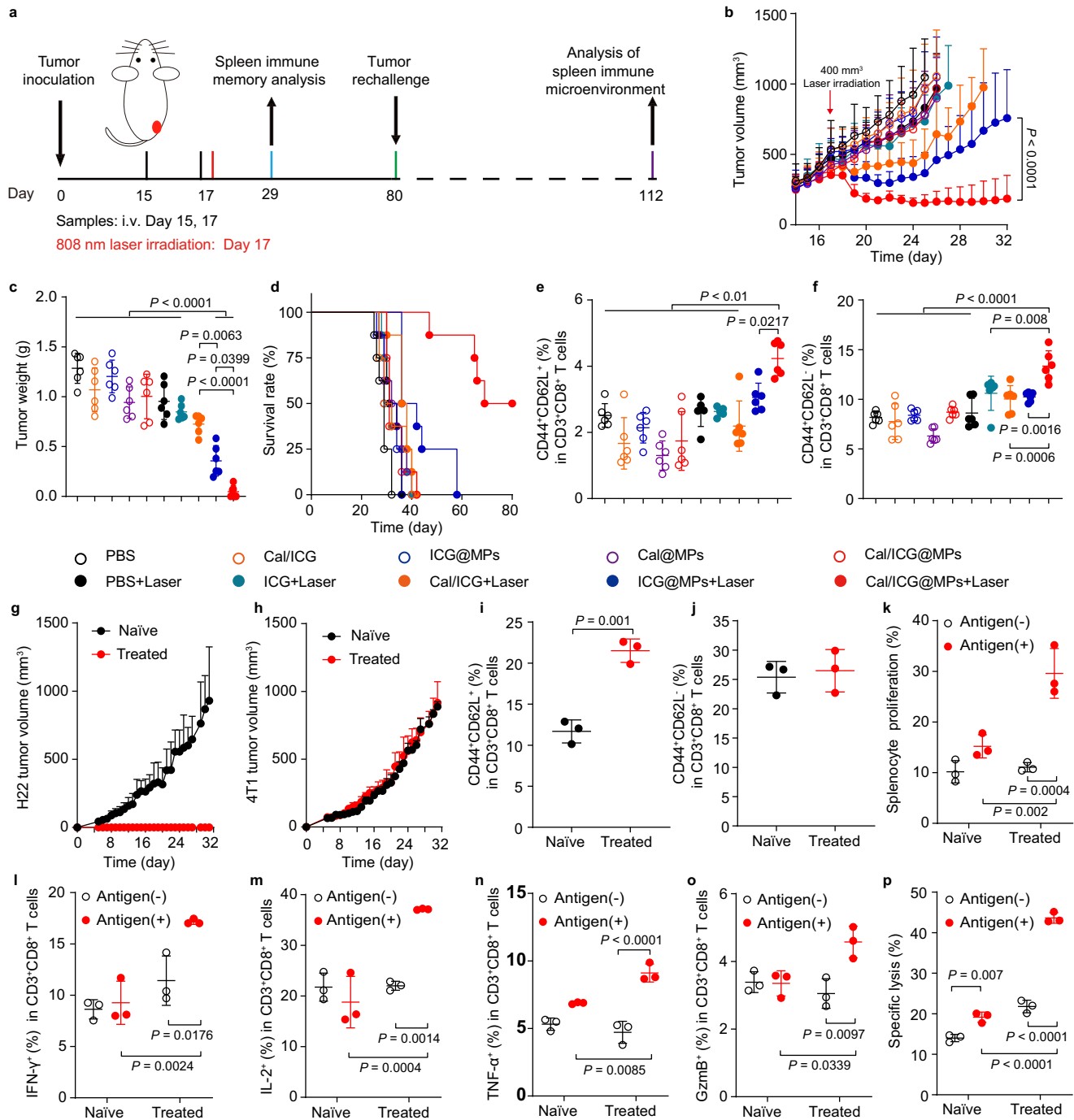

ICG@MPs with 808 nm laser irradiation-treated group (Fig. 7c). Kaplan–Meier survival analysis showed that 50% of mice were still alive in Cal/ICG@MPs with 808 nm laser irradiation-treated group after 80 d when all mice in other groups died (Fig. 7d). Hematoxylin-eosin (H&E) staining further confirmed the good antitumor activity of Cal/ICG@MPs upon 808 nm laser irradiation, as evidenced by the significant nucleus dissociation (Supplementary Fig. 37a). Meanwhile, the maximum proportion of TUNEL-positive apoptotic cells (Supplementary Fig. 37b) and lowest Ki67-positive proliferative tumor cells (Supplementary Fig. 37c) in tumor tissues were detected in Cal/ICG@MPs-treated group upon 808 nm laser irradiation. Cal/ICG@MPs-induced good anticancer activity was further confirmed in stroma-rich H22 tumor-bearing mice constructed by co-injection of H22 cells and HSCs upon

808 nm laser irradiation, even better than the combination of ICG@MPs and Cal-treated group (Supplementary Fig. 38a–c). However, no significant difference in anticancer activity was detected in ICG@MPs- and Cal/ICG@MPs-treated groups in stroma-poor H22 tumor-bearing mice upon 808 nm laser irradiation (Supplementary Fig. 39a–c), further confirming that Cal-triggered CAF regulation to reduce tumor ECM contributed to the anticancer activity of Cal/ICG@MPs upon laser irradiation. No apparent toxicity was detected in Cal/ICG@MPs with 808 nm laser irradiation-treated group, as indicated by body weight (Supplementary Fig. 40), H&E staining of major organs (Supplementary Fig. 41), and serological analysis (Supplementary Fig. 42). Importantly, a significant increase in the ratios of central memory T (Tcm) cells ($CD3^+CD8^+CD44^+CD62L^+$ cells, Fig. 7e and

**Fig. 7 Antitumor activity of Cal/ICG@MPs in H22 tumor-bearing mice upon 808 nm laser irradiation. a** Schematic schedule for the antitumor experiment in stroma-rich H22 tumor-bearing mice after intravenous injection of PBS, ICG, Cal/ICG, ICG@MPs, Cal@MPs or Cal/ICG@MPs derived from H22 cells at the ICG dosage of 8 mg kg$^{-1}$ and Cal dosage of 120 μg kg$^{-1}$, followed with or without 808 nm laser irradiation (1.5 W cm$^{-2}$, 10 min) at 2 h after the last injection. **b, c** Average tumor growth curves (**b**) and tumor weights (**c**) of stroma-rich H22 tumor-bearing mice after treatment indicated in **a**. Data are presented as means ± s.d. ($n = 8$ mice per group; two-way ANOVA followed by Tukey's multiple comparisons post-test for **b**; $n = 6$ mice per group; one-way ANOVA followed by Tukey's HSD post hoc test for **c**). **d** Kaplan–Meier survival plot of stroma-rich H22 tumor-bearing mice after treatment indicated in **a**. ($n = 8$ mice per group). **e, f** Percentages of CD8$^+$ Tcm (**e**) and CD8$^+$ Tem cells (**f**) in spleens of stroma-rich H22 tumor-bearing mice after treatment indicated in **a**. Data are presented as means ± s.d ($n = 6$ mice per group; one-way ANOVA followed by Tukey's HSD post hoc test). **g, h** Tumor growth curves after rechallenge with H22 cells ($2 \times 10^6$ cells, **g**) and 4T1 cells ($2 \times 10^6$ cells, **h**) in naive mice or Cal/ICG@MPs with 808 nm laser irradiation-cured mice indicated in **a**. Data are presented as means ± s.d. ($n = 3$ mice per group). **i, j** Percentages of CD8$^+$ Tcm cells (**i**) and CD8$^+$ Tem cells (**j**) in spleens of the above tumor-rechallenged mice. Data are presented as means ± s.d. ($n = 3$ mice per group; two-tailed unpaired $t$ test). **k** Proliferation ratios of splenocytes after the splenocytes from the above tumor-rechallenged mice were treated with or without H22 cell lysates (antigen) for 72 h. Data are presented as means ± s.d. ($n = 3$ mice per group; two-way ANOVA followed by Tukey's multiple comparisons post test). **l–o** Percentages of IFN-γ$^+$ (**l**), IL-2$^+$ (**m**), TNF-α$^+$ (**n**), and GzmB$^+$CD8$^+$ T cells (**o**) after the splenocytes from the above tumor-rechallenged mice were treated with or without H22 cell lysates for 72 h. Data are presented as means ± s.d. ($n = 3$ mice per group; two-way ANOVA followed by Tukey's multiple comparisons post test). **p** In vitro cytotoxicity of splenocytes against H22 cells after the splenocytes (effector cells) from the above tumor-rechallenged mice were pretreated with or without H22 cell lysates for 72 h and then incubated with H22 cells (target cells) at the effector/target ratio of 20:1 for 6 h. Data are presented as means ± s.d. ($n = 3$ mice per group; two-way ANOVA followed by Tukey's multiple comparisons post-test). Source data and exact $P$ values for **e** are provided as a Source Data file.

Supplementary Fig. 31b) and effector memory T (Tem) cells (CD3$^+$CD8$^+$CD44$^+$CD62L$^-$ cells, Fig. 7f and Supplementary Fig. 31b) was detected in spleens of Cal/ICG@MPs with 808 nm laser irradiation-treated group compared with other groups. When the mice with complete tumor ablation in Cal/ICG@MPs with 808 nm laser irradiation-treated group were rechallenged with H22 cells at 80 d after tumor inoculation, 100% of the mice rejected the tumor rechallenge up to 31 d, while continuous tumor growth was observed in naïve mice inoculated with the same numbers of H22 cells (Fig. 7g and Supplementary Fig. 43a). Meanwhile, 100% of tumor formation was observed in these mice with complete tumor ablation when inoculated with 4T1 cells (Fig. 7h and Supplementary Fig. 43b). These results suggest that Cal/ICG@MPs might inhibit tumor recurrence by generating immunological memory upon 808 nm laser irradiation.

To further confirm Cal/ICG@MPs with 808 nm laser irradiation-induced immunological memory, the ratios of Tcm and Tem cells in spleens of these rechallenged tumor-bearing mice were determined. Consistently, the higher ratio of Tcm cells (Fig. 7i), but not Tem cells (Fig. 7j) was detected in Cal/ICG@MPs with 808 nm laser irradiation-treated group. Furthermore, the splenocytes from these tumor-rechallenged mice were restimulated with H22 cell lysates for 3 days. The proliferation of splenocytes in Cal/ICG@MPs with 808 nm laser irradiation-treated group was significantly higher after restimulation of H22 tumor antigen (Fig. 7k). Meanwhile, the ratios of CD8$^+$IFN-γ$^+$ T (Fig. 7l), CD8$^+$ IL-2$^+$ T (Fig. 7m), CD8$^+$TNF-α$^+$ T (Fig. 7n), and CD8$^+$Granzyme B$^+$ T (Fig. 7o) cells in the splenocytes of Cal/ICG@MPs with 808 nm laser irradiation-treated mice was significantly increased after tumor antigen restimulation. When the splenocytes from these tumor-rechallenged mice were restimulated with H22 cell lysates for 3 days and then incubated with H22 cells, stronger cytotoxicity against H22 cells was detected in Cal/ICG@MPs with 808 nm laser irradiation-treated group (Fig. 7p). Taken together, these results confirm that Cal/ICG@MPs induce long-term immune memory to kill tumor cells and prevent tumor recurrence upon 808 nm laser irradiation.

**Cal/ICG@MPs exhibit excellent antitumor activity against bilateral 4T1 orthotopic tumors upon 808 nm laser irradiation.** To further evaluate the antitumor activity of Cal/ICG@MPs upon 808 nm laser irradiation, the highly malignant bilateral 4T1 orthotopic tumor models were constructed by co-injection of 4T1 cells and TGF-β-activated skin fibroblasts into the left (the

primary tumor) and right (contralateral tumor) breast fat pad on day 0 and 4, respectively. On day 9, the mice were intravenously injected with PBS, ICG, Cal/ICG, ICG@MPs, Cal@MPs, or Cal/ICG@MPs derived from 4T1 cells twice every 2 days, followed with or without 808 nm laser irradiation at the primary tumors for 10 min (Fig. 8a). The lowest tumor volume and weight of both primary (Fig. 8b, d) and contralateral tumors (Fig. 8c, e) were detected in Cal/ICG@MPs with 808 nm laser irradiation-treated group, leading to the longest survival time of about 60 days compared with other groups (Fig. 8f). These results suggest that Cal/ICG@MPs exhibit excellent antitumor activity against bilateral 4T1 orthotopic tumors upon 808 nm laser irradiation. Consistently, Cal/ICG@MPs significantly reduced primary tumor ECM as observed by Masson's trichrome staining (Supplementary Fig. 44a, b). Meanwhile, Cal/ICG@MPs significantly promoted DC maturation (Fig. 8g and Supplementary Fig. 45a) and tumor infiltration of CD8$^+$ T cells (Fig. 8h and Supplementary Fig. 45b), activated CD8$^+$CD69$^+$ T (Fig. 8i and Supplementary Fig. 45c), CD8$^+$IFN-γ$^+$ T (Fig. 8j and Supplementary Fig. 45d) and CD8$^+$ IL-2$^+$ T cells (Fig. 8k and Supplementary Fig. 45e), while decreased the ratios of Tregs (Fig. 8l and Supplementary Fig. 45f) and PMN-MDSCs (Fig. 8m and Supplementary Fig. 45g) in the contralateral tumors upon 808 nm laser irradiation. Similar phenomena were detected in the spleens of bilateral 4T1 orthotopic tumor-bearing mice (Supplementary Fig. 46a–g). Meanwhile, the highest ratio of Tcm cells (Supplementary Fig. 46h) and Tem cells (Supplementary Fig. 46i) was detected in spleens of Cal/ICG@MPs with 808 nm laser irradiation-treated group. These results suggest that Cal/ICG@MPs efficiently activate antitumor immunity to induce an abscopal effect upon 808 nm laser irradiation.

**Cal/ICG@MPs inhibit tumor metastasis in 4T1 metastasis tumor models upon 808 nm laser irradiation.** To further evaluate the anti-metastasis ability of Cal/ICG@MPs upon 808 nm laser irradiation, the orthotopic stroma-rich 4T1 tumor-bearing mice were intravenously injected with PBS, ICG, Cal/ICG, ICG@MPs, Cal@MPs, or Cal/ICG@MPs derived from 4T1 cells twice every 2 days, followed with or without 808 nm laser irradiation at the tumor sites for 10 min. During the process, the luciferase-expressing 4T1 (4T1-Luc) cells were intravenously injected to mimic the metastasis (Fig. 9a). Consistently, Cal/ICG@MPs significantly inhibited tumor growth (Supplementary Fig. 47a) and reduced tumor ECM (Supplementary 47b, c) of

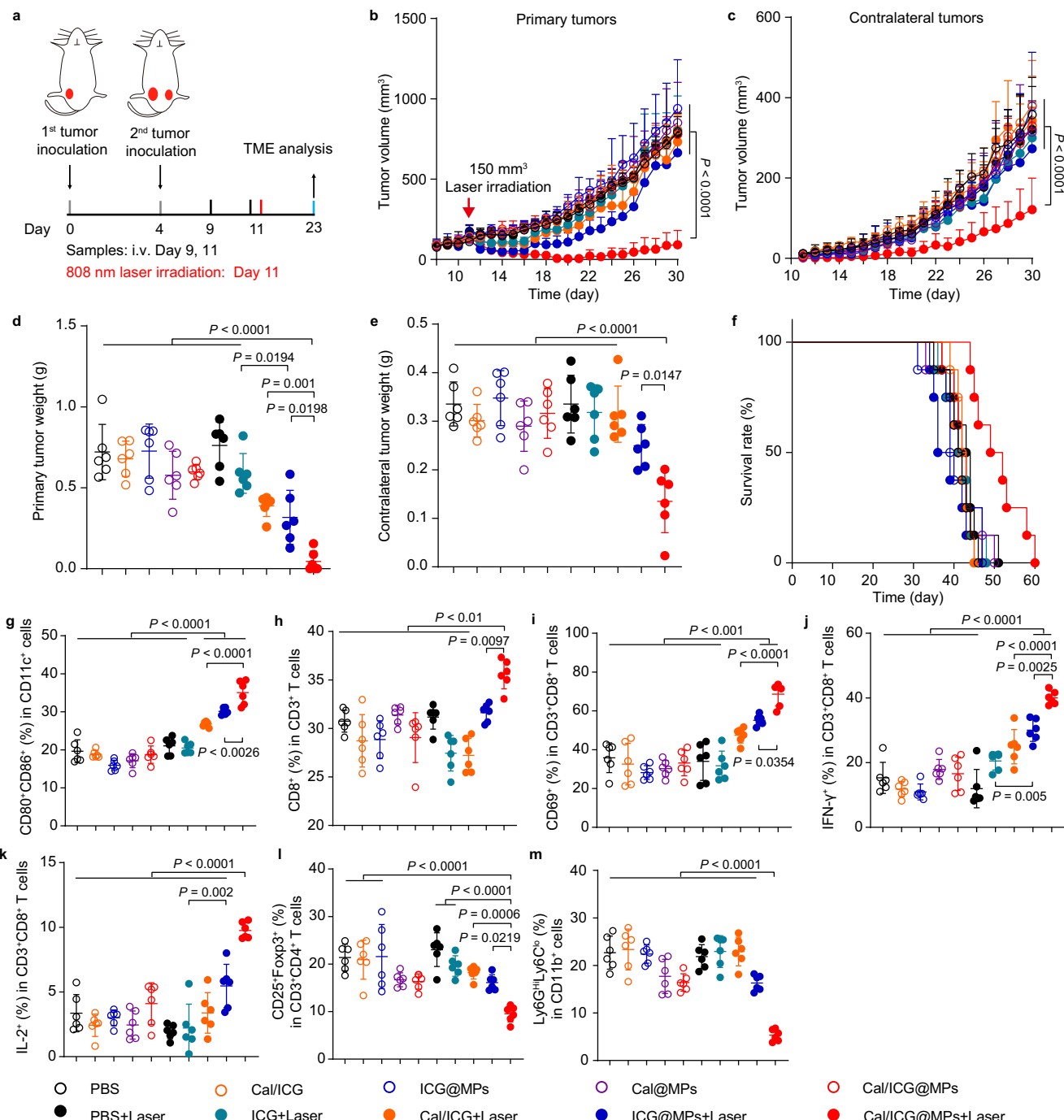

**Fig. 8 Cal/ICG@MPs-triggered inhibition in distant tumor growth in a 4T1 orthotopic bilateral tumor model upon 808 nm laser irradiation. a** Schematic illustration for the distant tumor inhibition experiment in a stroma-rich 4T1 orthotopic bilateral tumor model after intravenous injection of PBS, ICG, Cal/ICG, ICG@MPs, Cal@MPs, or Cal/ICG@MPs derived from 4T1 cells at the ICG dosage of 8 mg kg$^{-1}$ and Cal dosage of 120 µg kg$^{-1}$, followed with or without 808 nm laser irradiation (1.5 W cm$^{-2}$, 10 min) at 2 h after the last injection. **b, c** Primary (**b**) and contralateral (**c**) tumor growth curves of stroma-rich 4T1 orthotopic tumor-bearing mice after treatment indicated in **a**. Data are presented as means ± s.d. ($n = 6$ mice per group; two-way ANOVA followed by Tukey's multiple comparisons post-test). **d, e** Primary (**d**) and contralateral (**e**) tumor weights of stroma-rich 4T1 orthotopic tumor-bearing mice after treatment indicated in **a**. Data are presented as means ± s.d. ($n = 6$ mice per group; one-way ANOVA followed by Tukey's HSD post hoc test). **f** Kaplan–Meier survival plot of stroma-rich 4T1 orthotopic tumor-bearing mice after treatment indicated in (**a**). ($n = 8$ mice per group). **g–m** Percentages of DC maturation (**g**), CD8$^+$ T cells (**h**), CD8$^+$CD69$^+$ T cells (**i**), CD8$^+$IFN-γ$^+$ T cells (**j**), CD8$^+$IL-2$^+$ T cells (**k**), Tregs (**l**) and PMN-MDSCs (**m**) in the contralateral tumors of stroma-rich 4T1 orthotopic tumor-bearing mice after treatment indicated in **a**. Data are presented as means ± s.d. ($n = 6$ mice per group; one-way ANOVA followed by Tukey's HSD post hoc test). Source data and exact $P$ values for **h**, **i** are provided as a Source Data file.

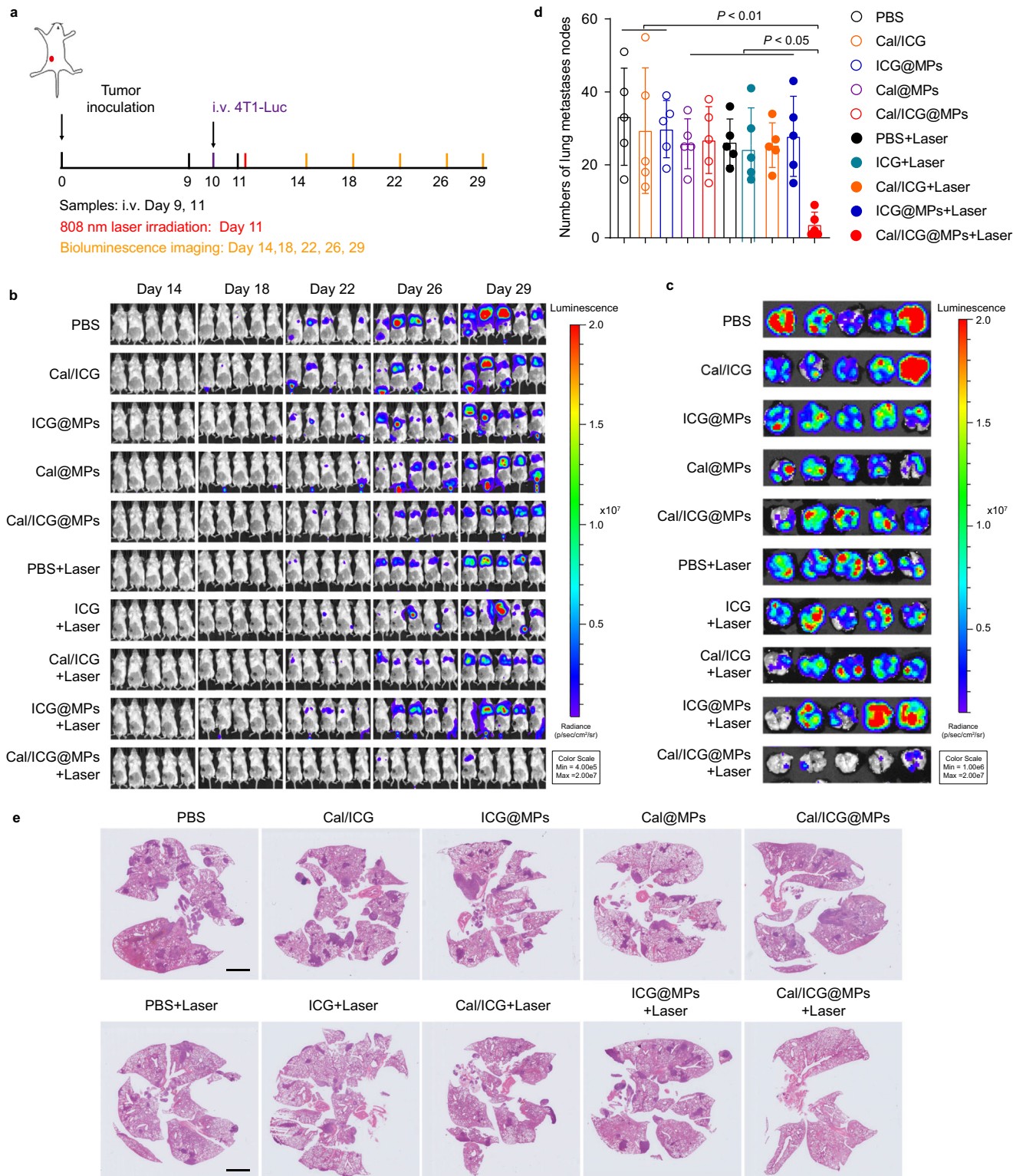

**Fig. 9 Cal/ICG@MPs-triggered metastasis inhibition upon 808 nm laser irradiation. a** Schematic illustration for the metastasis inhibition experiment in stroma-rich 4T1 orthotopic tumor-bearing mice after intravenous injection of PBS, ICG, Cal/ICG, ICG@MPs, Cal@MPs or Cal/ICG@MPs derived from 4T1 cells at the ICG dosage of 8 mg kg$^{-1}$ and Cal dosage of 120 μg kg$^{-1}$, followed with or without 808 nm laser irradiation (1.5 W cm$^{-2}$, 10 min) at 2 h after the last injection. **b** In vivo bioluminescence images tracking the spreading and growth of intravenously injected 4T1-Luc cells in stroma-rich 4T1 orthotopic tumor-bearing mice after treatment indicated in **a**. **c** Bioluminescence images of ex vivo lung tissues at the end of treatments indicated in **a**. **d** Numbers of lung nodules at the end of treatments indicated in **a**. Data are presented as means ± s.d. (n = 5 mice per group; one-way ANOVA followed by Tukey's HSD post hoc test). **e** H&E staining images of the lungs at the end of treatments indicated in **a**. Images are representative of 5 biologically independent mice. Scale bar: 2 mm. Source data and exact P values for **d** are provided as a Source Data file.

orthotopic 4T1 tumors upon 808 nm laser irradiation. In vivo bioluminescence imaging showed that the bioluminescence signals representing cancer metastasis in these mice became stronger with time (Fig. 9b). However, Cal/ICG@MPs-treated group exhibited negligible bioluminescence signals upon 808 nm laser irradiation (Fig. 9b). Meanwhile, ex vivo lung tissue images showed that the lowest bioluminescence intensity was detected in Cal/ICG@MPs with 808 nm laser irradiation-treated group (Fig. 9c), indicating that Cal/ICG@MPs significantly inhibited lung metastasis against the intravenously injected 4T1-Luc cells upon 808 nm laser irradiation. Furthermore, the significantly fewer metastatic nodules in the lungs were confirmed in Cal/ICG@MPs with 808 nm laser irradiation-treated group, as evidenced by both counting nodule numbers (Fig. 9d and Supplementary Fig. 48) and H&E staining (Fig. 9e). These results suggest that Cal/ICG@MPs exhibit good anti-metastasis effects upon 808 nm laser irradiation by activating antitumor immunity.

## Discussion

The insufficient tumor accumulation of photosensitizers not only weakens PTT-induced direct tumor cell killing but also compromises PTT-triggered antitumor immunity[9], limiting the therapeutic efficacy of PTT. Although tumor cell-derived MPs hold great potential as tumor-targeted drug-delivery carriers[30], it remains a big challenge for these MPs to significantly accumulate and penetrate into tumor tissues due to aberrant vascular architecture, elevated interstitial fluid pressure (IFP), and compact ECM in tumor tissues[5,6,46]. Meanwhile, these complex physiological and pathological barriers of tumor tissues restrict the tumor infiltration of PTT-activated CD8+ T cells[5,47].

CAFs exert diverse functions, including matrix deposition and remodeling, and extensive reciprocal signaling interactions with cancer cells and immune cells, to promote tumor progression, which makes them promising cancer therapeutic targets[11,12]. Nowadays, strategies regulating CAFs include direct CAF depletion and deactivation of CAFs to a quiescent state[22,24,48]. However, the lack of specific CAF markers makes it difficult to precisely target CAFs without damaging normal tissue, restraining their direct depletion. Meanwhile, deletion of stromal CAFs may not control tumor progression due to the complexity of CAF functions and possible interconvertibility of subtype[22,49]. For example, depletion of a-SMA+ myofibroblasts in pancreatic cancer resulted in the enhanced tumor invasion and increased ratio of Treg cells associated with decreased survival[22]. Therefore, attempts to regulate CAFs to normal fibroblasts could provide effective therapeutic modality in combination with other effective treatments such as PTT. In the present study, we developed tumor cell-derived MPs loading Cal and ICG (Cal/ICG@MPs) for synergistic combination therapy of CAF regulation and PTT. Cal/ICG@MPs efficiently targeted tumor tissues and decreased CAF numbers with the proliferation inhibition, showing a CAF-poor microenvironment following treatment. Here, Cal/ICG@MPs may not be actively depleting already present fibroblasts since no apoptosis is detected. However, whether Cal/ICG@MPs did not deplete CAFs needed to be further elucidated. Cal/ICG@MPs-triggered tumor ECM reduction significantly enhanced tumor accumulation and penetration of ICG, integrating with the tumor cell targeting of Cal/ICG@MPs to generate strong PTT efficacy against tumor cells. These data suggest that loading Cal and ICG to tumor cell-derived MPs fully acted on the photothermal and CAF regulation effects to achieve the synergistic treatment efficacy.

Tumor antigen-specific CD8+ CTL-mediated killing of tumor cells plays a crucial role in cancer immunotherapy[50,51]. Successful CTL-mediated tumor rejection requires the recruitment, infiltration, and expansion of tumor antigen-specific CTLs in tumor tissues[52,53]. CAFs have been shown to abrogate CD8+ T-cell function by restricting CTL recruitment to tumor tissues[44,54], producing aberrant ECM deposition to function as a physical barrier against tumor infiltration of CTLs[15], and driving antigen-mediated AICD of tumor-specific CTLs present within the tumor milieu by the apoptosis induction of CD8+ T cells via Fas (expressed in CD8+ T cells) and FasL (expressed in CAFs) pathway, as well as CD8+ T-cell anergy via PD-1 (expressed in CD8+ T cells) and PD-L2 (expressed in CAFs) pathway[19]. In this work, Cal/ICG@MPs-triggered PTT efficiently induced ICD effects of tumor cells, resulting in DC maturation and CD8+ T-cell activation, which mainly occurs in LNs. Cal/ICG@MPs-induced CAF regulation allowed the recruitment of the activated CD8+ T cells to tumor tissues, and meanwhile reduced tumor ECM to improve CD8+ T-cell infiltration to deep tumor parenchyma. Importantly, Cal/ICG@MPs-caused CAF regulation significantly decreased CAF-induced AICD of these CD8+ T cells when CAFs processed and cross-presented tumor antigens derived from the Cal/ICG@MPs-triggered PTT-induced ICD of tumor cells, as evidenced by the decreased expression of MHC-I, PD-L2 and FasL in CAFs after treatment with Cal/ICG@MPs upon 808 nm laser irradiation and the correspondingly enhanced CD8+ T-cell proliferation and decreased CD8+ T-cell apoptosis. Thus, Cal/ICG@MPs significantly increased CD8+ T-cell numbers in tumor tissues upon 808 nm laser irradiation compared with ICG@MPs with 808 nm laser irradiation-treated group. Moreover, Cal/ICG@MPs treatment significantly improved tumor-suppressive microenvironment upon 808 nm laser irradiation, which contributed to CTL-induced antitumor immune response. Cal/ICG@MPs-induced CAF regulation holds great potential to improve PTT-induced antitumor immunity and generate long-term immunological memory functions.

In summary, the data in this study show that Cal/ICG@MPs efficiently target tumor tissues and regulate CAFs to reduce tumor ECM, leading to not only the enhanced tumor accumulation and penetration of ICG to enhance PTT-induced CD8+ T cell-mediated antitumor immunity, but also the enhanced tumor infiltration of CD8+ T cells and ameliorated CAF-induced antigen-mediated AICD of CD8+ T cells in response to PTT with the reshaped tumor immune-suppressive microenvironment. With these features, Cal/ICG@MPs efficiently inhibit tumor recurrence and metastasis upon 808 nm laser irradiation by inducing a long-term antitumor immune memory. These findings reveal that Cal/ICG@MPs function as a promising platform to achieve better therapeutic effects in tumor PTT.

## Methods

**Materials**. RPMI 1640 medium, Dulbecco's Modified Eagle's Medium (DMEM), FBS, and collagenase Type I were purchased from Gibco BRL/Life Technologies (Grand Island, NY, USA). GM-CSF, IL-4, IL-2, and TGF-β were provided by PeproTech (Rocky Hill, NJ, USA). ICG was obtained from J&K China Chemical Ltd. (Beijing, China). Calcipotriol hydrate and AOM were purchased from Sigma-Aldrich (St Louis, MO, USA). DSS was purchased from MP Biomedicals (Santa Ana, CA, USA). Antibodies used for flow cytometric analysis were purchased from BioLegend (San Diego, CA, USA). All other reagents were of analytical grade and used without any further purification.

**Cell culture**. H22, 4T1, and HepG2 cells were obtained from Type Culture Collection of Chinese Academy of Sciences (Shanghai, China). HSCs were kindly provided by Prof. Guangjun Nie's group (National Center for Nanoscience and Technology, Beijing, China). MC38 cells were bought from American Type Culture Collection (ATCC). Primary human lung fibroblasts were kindly provided by Dr. Qing Zhou (Tongji Hospital, Huazhong University of Science and Technology, Wuhan, China). BMDCs were generated by isolating bone marrow cells from 8-week-old male BALB/c mice and culturing them in a medium containing 10 ng mL$^{-1}$ GM-CSF and 10 ng mL$^{-1}$ IL-4 for 7 days[28]. Skin fibroblasts were isolated by outgrowth using explant techniques from the skin of suckling mice[55]. Briefly, small skin blocks were cut (0.5–1 mm$^3$) using a razor blade and seeded in uncoated culture flasks. The culture medium was changed every

2 days for three times. Fibroblasts grew out from the tissue blocks 1–7 days later and then were induced with 40 ng mL$^{-1}$ TGF-β for 24 h to myofibroblasts. For evaluation of TGF-β-induced myofibroblast, the expression of α-SMA and FAP in myofibroblasts was determined by western blot. The following primary antibodies were used: mouse anti-β-actin (Proteintech, cat. No 60008-1-1 g, clone 7D2C10, 1/2000 dilution), anti-α-SMA (Novus, cat No NBP2-33006, clone 1A4/asm-1, 1/1000 dilution) and anti-FAP (Novus, cat. No NB110-85534, 1/1000 dilution). Myofibroblasts between passages 3 and 8 were used for the following experiments. Splenocytes were collected from the 8-week-old male BALB/c mice and activated with anti-mouse CD3 antibody (Biolegend, cat. No 100340, clone 145-2C11, 1/200 dilution) and anti-mouse CD28 antibody (Biolegend, cat. No 102116, clone 37.51, 1/1000 dilution) for 2 days. CD8$^+$ T cells were isolated using a MACS CD8$^+$ T-cell isolation kit (Miltenyi Biotec, Germany) and expanded in a medium containing 20 ng mL$^{-1}$ IL-2 for another 5 days. CD3$^+$ T cells were isolated from the splenocytes of 8-week-old male BALB/c mice using a MojoSort$^{TM}$ Mouse CD3$^+$ T-cell isolation kit (Biolegend, USA) and expanded in a medium containing 20 ng mL$^{-1}$ IL-2 for 5 days. H22, 4T1 and MC38 cells, BMDCs, CD8$^+$ T and CD3$^+$ T cells were cultured in RPMI 1640 medium at 37 °C in a 5% CO$_2$ humidified incubator. Myofibroblasts and HepG2 cells were cultured in DMEM medium, and HSCs were cultured in DMEM/F12 medium. All media contained 10% FBS, 100 U mL$^{-1}$ penicillin, and 100 μg mL$^{-1}$ streptomycin.

**Human organotypic tumor slice culture**. Fresh HCC tissues were obtained from liver cancer patients experiencing routine surgical removal at the Tongji Hospital, Tongji Medical College of Huazhong University of Science and Technology (Wuhan, China). Tumors were cut into thin slices of about 5 × 5 × 5 mm and cultured in a complete RPMI 1640 medium. The study was approved by the Clinical Trial Ethics Committee of Huazhong University of Science and Technology and all patients had informed consent.

**Animal models**. BALB/c mice (male and female, 6–8-week-old) and C57BL/6 mice (male, 8-week-old) were purchased from Beijing Vital River Laboratory Animal Technology Co., Ltd. (Beijing, China). Mice were housed in an animal facility under constant environmental conditions (room temperature, 21 ± 1 °C; relative humidity, 40 − 70%; a 12 h light-dark cycle). All mice had access to food and water ad libitum. To establish a stroma-rich H22 subcutaneous liver cancer model, $2 \times 10^6$ H22 cells and $1 \times 10^6$ CAFs (HSCs or TGF-β-activated skin fibroblasts) were subcutaneously injected into the flanks of male BALB/c mice. To establish a rechallenge tumor model, stroma-rich H22 tumor-bearing mice cured by Cal/ICG@MPs with 808 nm laser irradiation were subcutaneously injected with $2 \times 10^6$ H22 cells and $1 \times 10^6$ CAFs into the left flank, and $2 \times 10^6$ 4T1 cells and $1 \times 10^6$ CAFs into the right flank. To establish a stroma-rich 4T1 orthotopic breast cancer model, $5 \times 10^5$ 4T1 cells and $2.5 \times 10^5$ CAFs were inoculated into the left fourth breast fat pad of BALB/c female mice. Four days later, $1 \times 10^5$ 4T1 cells and $5 \times 10^4$ CAFs were inoculated into the right fourth breast fat pad of the mice to establish the bilateral tumor model. To establish the lung metastasis model, $1 \times 10^5$ luciferase-transfected 4T1 (4T1-Luc) cells were intravenously administered via tail vein. AOM/DSS-induced CAC mouse model was constructed by intraperitoneally injecting male C57BL/6 mice with 10 mg/kg AOM, followed by receiving 3 cycles of 2.5% DSS in drinking water for 1 week and then drinking regular water for 2 weeks. All animal experiments were performed under the guidance and approved by the Institutional Animal Care and Use Committee at Tongji Medical College, Huazhong University of Science and Technology (Wuhan, China).

**Preparation and characterization of drug-loaded MPs**. H22, 4T1, HepG2, or MC38 cells were subjected to ultraviolet irradiation (UVB; 300 J m$^{-2}$) for 1 h, and then treated with 100 μg mL$^{-1}$ ICG or/and 4 μg mL$^{-1}$ Cal. After 12 h incubation, the supernatants were first centrifuged at 600 × g for 10 min and 18,000 × g for 2 min to remove the debris. The supernatants were further centrifuged at 18,000 × g for 30 min to harvest drug-loaded MPs. The pellets were washed three times with PBS and then resuspended in a cell culture medium for the following experiments. The concentration of ICG in MPs was measured using a UV spectrophotometer at 780 nm. The concentration of Cal in MPs was measured by HPLC. Chromatography was performed on a Hypersil ODS-2 column (250 mm × 4.6 mm; 5 μm), in which the mobile phases were 0.01 mol L$^{-1}$ diammonium hydrogen phosphate solution-methanol-acetonitrile (30: 30: 40). The effluents were monitored at 268 nm. The hydrodynamic diameters and zeta potentials of drug-loaded MPs were measured by DLS (Zetasizer Nano ZS90, Malvern Instruments, Worcestershire, UK). Their morphology was observed by TEM (Tecnai G2-20, FEI, Netherlands) with an accelerating voltage of 80 kV.

**In vitro photothermal effects**. ICG, Cal/ICG, ICG@MPs, or Cal/ICG@MPs at different ICG concentrations were irradiated with 808 nm laser at a power of 1.0 W cm$^{-2}$ for different time intervals. The temperature of the solutions was monitored by an FLIR E50 Infrared (IR) camera (FLIR Systems Inc., Wilsonville, OR, USA). PBS was used as a negative control in this experiment.

**Cellular uptake**. H22 or HepG2 cells were treated with ICG, Cal/ICG, ICG@MPs, or Cal/ICG@MPs at the ICG concentration of 0.5 μg mL$^{-1}$ and Cal concentration of 7 ng mL$^{-1}$ for different time intervals. The intracellular ICG fluorescence was

detected by flow cytometry (CytoFLEX S, Beckman Coulter, Fullerton, CA, USA) and analyzed by CytExpert (Beckman Coulter) software.

**Cell viability assay**. H22, HepG2, 4T1 cells, or myofibroblasts were treated with PBS, ICG, Cal/ICG, ICG@MPs, Cal@MPs or Cal/ICG@MPs at different ICG concentrations for 4 h. The cells were then washed twice with PBS and irradiated with or without 808 nm laser at a power of 1.0 W cm$^{-2}$ for 5 min. After 4 h incubation, the cell viability was measured using a cell counting kit-8 (CCK-8) assay. Briefly, 10 μL of CCK-8 solution (Dojindo, Kumamoto, Kyushu, Japan) was added to each well. After incubation for 4 h, the absorbance at 450 nm was measured on a MultiSkan FC microplate reader (Thermo Scientific, Waltham, MA, USA).

**In vitro ICD detection**. H22, HepG2, or 4T1 cells were treated with PBS, ICG, Cal/ICG, ICG@MPs, Cal@MPs, or Cal/ICG@MPs at the ICG concentration of 4 μg mL$^{-1}$ and Cal concentration of 60 ng mL$^{-1}$ for 4 h. The cells were then irradiated with or without 808 nm laser at a power of 1.0 W cm$^{-2}$ for 5 min. After 4 h incubation, the cells were washed twice with PBS and then incubated with an anti-CRT antibody (R&D Systems, USA, cat. No MAB38981, clone 681233, 2.5 μg mL$^{-1}$) for 1 h at 4 °C. Subsequently, the cells were further washed twice with PBS and incubated with FITC-conjugated goat anti-mouse IgG (Servicebio, China, cat. No GB22301,1/100 dilution) for 1 h. The expression of CRT was detected using confocal microscopy (FV1000, Olympus) and flow cytometry. The extracellular release of HMGB1 and ATP was examined using the HMGB1 ELISA kit (Moshake, Wuhan, China) and chemiluminescence ATP determination kit (Beyotime, Shanghai, China) according to the manufacturer's instructions, respectively.

**In vitro BMDC maturation and T-cell activation**. H22 or 4T1 cell were treated with PBS, ICG, Cal/ICG, ICG@MPs, Cal@MPs, or Cal/ICG@MPs at the ICG concentration of 4 μg mL$^{-1}$ and Cal concentration of 60 ng mL$^{-1}$ for 4 h. The cells were then irradiated with or without 808 nm laser at a power of 1.0 W cm$^{-2}$ for 5 min. After 12 h incubation, the tumor cell supernatants were collected and added to immature BMDCs. After 24 h incubation, BMDCs were stained with fluorescence-labeled anti-CD11c (Biolegend, cat. No 117328, clone N418, 1/20 dilution), CD80 (Biolegend, cat. No 104708, clone 16-10-A1, 1/40 dilution) and CD86 (Biolegend, cat. No 105012, clone GL-1, 1/80 dilution), and then detected by flow cytometry and analyzed by Flowjo (BD Biosciences)[56].

To assess the activation levels of T lymphocytes[57], CD3$^+$ T cells were co-cultured with the above-matured BMDCs at a BMDC/T-cell ratio of 1:10. After 5-day incubation, the CD3$^+$ T cells were washed with PBS three times and subsequently stimulated with a cell activation cocktail (with Brefeldin A, Biolegend, cat. No 423303) for 2 h. The cells were stained with anti-mouse CD8-PE/Cy7 antibody (Biolegend, cat. No 100722, clone 53–6.7, 1/80 dilution), and then fixed with intracellular staining fixation buffer (Biolegend, cat. No 420801) and permeabilized with permeabilization wash buffer (Biolegend, cat. No 421002) according to the manufacturer's instructions. The cells were further stained with fluorescence-labeled anti-IL-2 (Biolegend, cat. No 503808, clone JES6-5H4, 1/20 dilution), Granzyme B (Biolegend, cat. No 372204, clone QA16A02, 1/20 dilution), IFN-γ (Biolegend, cat. No 505810, clone XMG1.2, 1/80 dilution) or TNF-α (Biolegend, cat. No 506306, clone MP6-XT22, 1/80 dilution) for 30 min at room temperature. The cells were washed with PBS and then subjected to flow cytometric analysis.

**In vitro CAF regulation and ECM remodeling**. Myofibroblasts or human organotypic tumor slices from liver cancer patients were treated with PBS, ICG, Cal, Cal/ICG, ICG@MPs, Cal@MPs, or Cal/ICG@MPs at the ICG concentration of 4 μg mL$^{-1}$ and Cal concentration of 60 ng mL$^{-1}$ for 48 h incubation. For evaluation of Cal/ICG@MPs-induced myofibroblast regulation, the expression of α-SMA and fibronectin in myofibroblasts was determined by western blot and immunofluorescence analysis, and the collagen deposition was detected by Sirius red total collagen detection kit (Chondrex, Redmond, WA, USA) according to the manufacturer's instructions. For evaluation of Cal/ICG@MPs-induced CAF quiescence and ECM reduction in human organotypic tumor slices, the tumor slices were washed three times with PBS, then fixed with 4% paraformaldehyde, sectioned and examined by Masson's trichrome staining, and immunofluorescence staining for α-SMA, collagen-I, and fibronectin. The following primary antibodies were used: rabbit anti-tubulin (CST, cat No 86298 T, clone D3U1W, 1/1000 dilution), rabbit anti-fibronectin (Proteintech, cat No 15613-1-AP, 1/1000 dilution for western blot, 1/200 dilution for immunofluorescence), mouse anti-α-SMA (NOVUS, cat No NBP2-33006, clone 1A4/asm-1, 1/1000 dilution for western blot, 1/200 dilution for immunofluorescence) and rabbit anti-collagen-I (NOVUS, cat No NB600-408, 1/500 dilution).

**In vivo photothermal effects**. When the tumor volume of stroma-rich H22 tumor-bearing mice and 4T1 tumor-bearing mice reached about 300 mm³ or 100 mm³, respectively, the mice were intravenously injected with PBS, ICG, Cal/ICG, ICG@MPs, or Cal/ICG@MPs at the ICG dosage of 8 mg kg$^{-1}$ and Cal dosage of 120 μg kg$^{-1}$ twice every 2 days. At 2 h after the last injection, the mice were anesthetized and tumors were irradiated with 808 nm laser at a power of 1.5 W cm$^{-2}$ for 10 min. The thermal images and region maximum temperature of the tumors were recorded by the IR thermal camera.

**Ex vivo tumor penetration**. The tumors of stroma-rich H22 tumor-bearing mice (~300 mm$^3$) or human organotypic tumor slices from liver cancer patients were incubated in fresh RPMI 1640 media containing ICG, Cal/ICG, ICG@MPs, or Cal/ICG@MPs at the ICG concentration of 4 μg mL$^{-1}$ and Cal concentration of 60 ng mL$^{-1}$. At 48 h after incubation, the tumors were washed three times with PBS, then frozen sectioned, and examined using confocal microscopy.

**Chemotaxis assay**. A chemotaxis assay was performed using 5-μm transwell filter (Corning Costar, Acton, MA, USA). Myofibroblasts were treated with PBS, ICG, Cal, Cal/ICG, ICG@MPs, Cal@MPs or Cal/ICG@MPs at the ICG concentration of 4 μg mL$^{-1}$ and Cal concentration of 60 ng mL$^{-1}$. After 48 h incubation, the supernatants were collected as conditional media and added to the bottom chambers. CD3$^+$ T cells (1 × 10$^5$ cells) were seeded in the top chambers. After 8 h, the suspensions from the bottom chambers were collected and CD3$^+$CD8$^+$ T cells were detected by flow cytometry.

**Myofibroblast-induced antigen-mediated AICD effects of CD8$^+$ T cells**. H22 or 4T1 cells were treated with PBS, ICG, Cal/ICG, ICG@MPs, Cal@MPs, or Cal/ICG@MPs at the ICG concentration of 4 μg mL$^{-1}$ and Cal concentration of 60 ng mL$^{-1}$ for 4 h. The cells were then irradiated with or without 808 nm laser at a power of 1.0 W cm$^{-2}$ for 5 min. At 12 h after incubation, the supernatants were collected as conditioned media. Myofibroblasts were seeded on six-well plates at a density of 2.0 × 10$^5$ cells well$^{-1}$ and treated with PBS, ICG, Cal/ICG, ICG@MPs, Cal@MPs, or Cal/ICG@MPs at the ICG concentration of 4 μg mL$^{-1}$ and Cal concentration of 60 ng mL$^{-1}$ for 48 h. The treated myofibroblasts were then incubated with the above-conditioned media for 24 h and washed thoroughly with PBS three times. 2 × 10$^6$ CD8$^+$ T cells were then co-cultured with the above-treated myofibroblasts for 48 h. Myofibroblasts and CD8$^+$ T cells were collected for flow cytometric analysis. For determination of myofibroblast phenotyping, the cells were stained with FITC-anti-mouse H-2Kd/H-2Dd antibody (Biolegend, cat. No 114706, clone 34-1-2 S, 1/200 dilution), PE-anti-mouse CD178 (FasL) antibody (Biolegend, cat. No 106605, clone MFL3, 1/80 dilution) or APC-anti-mouse CD273 (PD-L2) antibody (Biolegend, cat. No 107210, clone TY25, 1/80 dilution). For determination of CD8$^+$ T-cell proliferation, CD8$^+$ T cells were stained with CFSE (Invitrogen, C34570) before co-culture. For determination of CD8$^+$ T-cell apoptosis, CD8$^+$ T cells were detected using Annexin V-FITC/PI apoptosis detection kit (Yeasen, Shanghai, China) according to the manufacturer's instructions.

**In vivo antitumor effect**. When tumor volume of H22 or 4T1 tumor-bearing mice reached around 300 mm$^3$ or 100 mm$^3$, respectively, the mice were divided into ten groups randomly and intravenously injected with PBS, ICG, Cal/ICG, ICG@MPs, Cal@MPs, or Cal/ICG@MPs at the ICG dosage of 8 mg kg$^{-1}$ and Cal dosage of 120 μg kg$^{-1}$ twice every 2 days. At 2 h after the last injection, the mice were irradiated with or without 808 nm laser at 1.5 W cm$^{-2}$ for 10 min. The tumor sizes were measured every day via Vernier Caliper and the body weight of mice were also recorded. The sizes of tumors were calculated as the following formula: width$^2$ × length × 0.5. For animal welfare, the mice were killed when the tumor volume grew to 1500 mm$^3$. On day 14 after treatment, one part of the mice was sacrificed, and tumors and major organs (heart, liver, spleen, lung, and kidney) were obtained. The tumors were weighed, then fixed with 4% paraformaldehyde, sectioned, and examined by H&E staining, TUNEL staining, Masson's trichrome staining, and immunofluorescence staining for α-SMA, collagen-I, and fibronectin. The major organs were also fixed and sectioned for H&E staining. The rest mice were used for long-term tumor inhibition and survival observation. For lung metastasis analysis, lung metastases were monitored via bioluminescence using the Caliper IVIS Lumina II in vivo imaging system (PerkinElmer, Waltham, MA, USA). Bioluminescence imaging was performed every 4 days to monitor the progression of tumor growth. At the end of this experiment, lungs were collected and fixed in Bouin's solution. The metastatic tumor nodules in the lungs were counted, then fixed, and sectioned for H&E staining.

**Tumor immune microenvironment analysis**. After sacrificing the tumor-bearing mice, the tumors, tumor-draining LNs, and spleens were excised. The tumors were cut into small pieces and then incubated with RPMI 1640 media containing 0.8 mg mL$^{-1}$ collagenase type I and 5 μg mL$^{-1}$ DNase I for 1 h at 37 °C with persistent agitation. The cells were collected by centrifugation at 300 × $g$ for 5 min, washed twice with PBS, and then filtered twice using a 40-μm filter. The red blood cells were lysed with red cells lysis solution (Biosharp, Hefei, China). The lymphocytes from tumor-draining LNs and spleens were obtained by mashing the tissues through a 40-μm filter and then lysing the red blood cells. For CAF analysis, cells were labeled with fluorescence-labeled anti-CD45 (Biolegend, cat. No 103108, clone 30-F11, 1/200 dilution; or cat. No 103138, clone 30-F11, 1/40 dilution), CD31 (Biolegend, cat. No 102424, clone 390, 1/40 dilution; or cat. No 102406, clone 390, 1/50 dilution), EpCAM (Biolegend, cat. No 118225, clone G8.8, 1/80 dilution), CD140α (Biolegend, cat. No 135916, clone APA5, 1/40 dilution), podoplanin (Biolegend, cat. No 127412, clone 8.1.1, 1/20 dilution), CD178 (Biolegend, cat. No 106605, clone MFL3, 1/80 dilution), CD273 (Biolegend, cat. No 107210, clone TY25, 1/80 dilution), H-2Kd/H-2Dd (Biolegend, cat. No 114716, clone 34-1-2 S, 1/40 dilution) or α-SMA (1A4/asm-1) (Novus, cat. No NBP2-33006AF532, clone 1A4/asm-1, 1/100 dilution). For Ki67 staining, the cells were treated with 70% ethanol after surface staining and re-stained with fluorescence-labeled

anti-Ki67 (Biolegend, cat. No 652406, clone 16A8, 1/80 dilution). For PMN-MDSC analysis, the cells were stained with fluorescence-labeled anti-CD11b (Biolegend, cat. No 101228, clone M1/70, 1/80 dilution), Ly-6G (Biolegend, cat. No 127607, clone 1A8, 1/80 dilution) and Ly-6C (Biolegend, cat. No 128016, clone HK1.4, 1/80 dilution). For lymphocyte analysis, tumor-infiltrating lymphocytes were isolated by Ficoll-Paque PLUS density gradient media (GE Healthcare, Piscataway, NJ, USA). For surface marker analysis, the cells were stained with fluorescence-labeled anti-CD11c (Biolegend, cat. No 117328, clone N418, 1/20 dilution), F4/80 (Biolegend, cat. No 123114, clone BM8, 1/80 dilution), CD80 (Biolegend, cat. No 104708, clone 16-10-A1, 1/40 dilution), CD86 (Biolegend, cat. No 105012, clone GL-1, 1/80 dilution), CD45 (Biolegend, cat. No 103108, clone 30-F11, 1/200 dilution), CD3 (Biolegend, cat. No 100218, clone 17A2, 1/20 dilution), CD4 (Biolegend, cat. No 100528, clone RM4-5, 1/80 dilution), CD8 (Biolegend, cat. No 100712, clone 53–6.7, 1/80 dilution), CD69 (Biolegend, cat. No 104545, clone H1.2F3, 1/80 dilution) or CD25 (Biolegend, cat. No 101910, clone 3C7, 1/80 dilution). For intracellular cytokine staining, the cells were then treated with Fix/Perm solution (Biolegend, cat. No 420801) and re-stained with fluorescence-labeled anti-IFN-γ (Biolegend, cat. No 505808, clone XMG1.2, 1/80 dilution), IL-2 (Biolegend, cat. No 503832, clone JES6-5H4,1/160 dilution). For transcription factor staining, the cells were treated with True-Nuclear™ Transcription Factor Buffer Set (Biolegend, cat. No 424401) after surface staining and re-stained with fluorescence-labeled anti-Foxp3 (Biolegend, cat. No 126404, clone MF-14, 1/20 dilution). Memory T cells in spleens were analyzed by staining with fluorescence-labeled anti-CD3 (Biolegend, cat. No 100218, clone 17A2, 1/20 dilution), CD8 (Biolegend, cat. No 100712, clone 53–6.7, 1/80 dilution), CD44 (Biolegend, cat. No 103005, clone IM7, 1/200 dilution), CD62L (Biolegend, cat. No 104418, clone MEL-14, 1/80 dilution). The antibodies were incubated with the cells for 30 min at room temperature. The cells were analyzed by the CytoFLEX S flow cytometry.

**Image acquisition and analysis of CD8$^+$ T cells**. Immunofluorescence images of CD8$^+$ T cells (labeling with Cy3-conjugated CD8 antibody, green) and CAFs (labeling with Cy5-conjugated α-SMA antibody, red) in tumor tissues of stroma-rich H22 tumor-bearing mice were acquired using Pannoramic Scanner (3DHISTECH, Hungary) and analyzed using CaseViewer (3DHISTECH, Hungary) and StrataQuest tissue analysis software (TissueGnostics, Asia Pacific limited, China) as described previously[58]. Briefly, the algorithm detected the nuclei on the basis of the signal from the DAPI channel, then expanded and built a mask over the cytoplasm. On the generated mask, the algorithm searched for the localization of α-SMA and CD8 signals. Results were plotted onto scattergrams or histograms, and events were manually verified for all quadrants. The numbers of CD8$^+$ T-cell distributed around CAFs were quantified using a distance transform engine by counting CD8$^+$ T cells <50 μm distant from CAFs.

**Cytokine assay**. Serum samples were obtained from H22 tumor-bearing mice after treatments. After centrifugation at 10,000 × $g$ for 10 min, the serum was collected for the detection of TNF-α and IFN-γ activity using ELISA kits according to the manufacturer's instructions (DAKEWE, China).

**Statistical analysis**. Experiments were repeated at least three times. Statistical analysis was performed using GraphPad Prism 7.0 software (GraphPad Software, CA). For comparison of multiple groups, one-way analysis of variane (ANOVA) or two-way ANOVA was used, followed by Tukey's honest significant difference post hoc test or Bonferroni's multiple comparisons post-test, except where otherwise noted. Comparison between the two groups was performed using an unpaired two-tailed Student's $t$ test. Results are expressed as means ± SD. $P < 0.05$ was considered statistically significant.

**Reporting summary**. Further information on research design is available in the Nature Research Reporting Summary linked to this article.

## Data availability
The main data supporting the results in this study are available within the Article, Supplementary Information, or Source Data File. Source data are provided in this paper.

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

## Acknowledgements

This work was supported by the National Basic Research Program of China (2020YFA0710700 and 2018YFA0208900 to L.G.), National Natural Science Foundation of China (81974459 to L.G., 82073796 and 81627901 to X.Y.), China Postdoctoral Science Foundation (2020T130228 to T.Y.) and Program for HUST Academic Frontier Youth Team (2018QYTD01 to L.G.). We thank the Research Core Facilities for Life Science (HUST), the

Analytical and Testing Center of Huazhong University of Science and Technology, the Optical Bioimaging Core Facility of WNLO-HUST, and Wuhan Institute of biotechnology for related analysis. We thank TissueGnostics Asia Pacific Limited for the related data analysis.

## Author contributions

L.G., T.Y., and X.Y. designed the project. X.L., T.Y., Z.W., N.B., X.Z., G.Z., J.L., J.Q., and J.Y. performed the experiments. X.L., T.Y., B.Z., L.G., and X.Y. analyzed and interpreted the data, and wrote the manuscript.

## Competing interests

The authors declare no competing interests.
