## [Peer Review File · Nature Communications]

Reviewers' Comments:

Reviewer #1:

Remarks to the Author:

In this manuscript, the authors designed a cancer cell-derived microparticles co-delivering calcipotriol and Indocyanine green (Cal/ICG@MPs) to deactivate CAFs for improved PTT efficacy. With the enhanced accumulation and distribution of the ICG in the tumor, a strong PTT efficacy and activation of CD8+ T cell-mediated antitumor immunity were observed. In addition, the authors also showed the long-term antitumor immune memory to inhibit tumor recurrence and metastasis via Cal/ICG@MPs. Overall, it was well organized and written. My concerns are listed below.

1. It is challenging to maintain the stability of the cell-derived microparticles in the aqueous solution. The authors demonstrated no significant size and zeta potential change of Cal/ICG@MPs over 7 days. Please give the explanation why these particles were stable.
2. The results supported Cal/ICG@MPs as a promising drug to improve PTT efficacy in cancer treatment. Did the authors try the ICG@MPs and Cal combination formulation? Whether it is better to treat the mice with Cal and followed by ICG@MPs?
3. Due to the different functions and targeted cells of the Cal and ICG, the rationale of combining the Cal and ICG in the microparticles should be discussed.
4. Although the ICG and Cal were loaded in the microparticles, whether the ratio of ICG and Cal in the manuscript was the best one. In addition, the drug loading and release mechanism should be studied.
5. Stroma-rich H22 tumor-bearing mice were utilized for the in vivo study. Whether Cal/ICG@MPs could accumulate in the CAFs? What's the PTT effect to this type of cells?

Reviewer #2:

Remarks to the Author:

Li et al have investigated means to increase the effectiveness of photothermal therapy by delivering the substance indocyanine green to tumor cells by tumor derived microparticles conjugated with calcipotriol. They show that these Cal/ICG@MPs effectively target the tumor, remodel the tumor microenvironment by deactivating CAFs, increase tumor-specific immunity and inhibit tumor growth. This is very interesting concept and the results look promising but there are some concerns that need to be addressed.

CAFs were generated from healthy murine skin explants and cultured in TGF-beta. This seems a bit simplistic since there are many other factors in the tumor microenvironment that can drive the differentiation of CAFs. Fibroblasts isolated from skin may also inherently be very different compared to fibroblasts isolated from other tissues. For instance, the majority of fibroblasts in the liver are stellate cells which can react differently compared to skin-derived fibroblasts in the presence of tumor cells. It has also recently been shown that there are several subtypes of CAFs with diverse functions and the subgroup expressing high levels of α -SMA (myCAFs) appears to protect from tumor invasion rather promote malignancy (Öhlund et al JEM, 2017 Biffi et al Cancer Discov 2019, Kieffer et al Cancer Discov 2020, Costa et al Cancer Cell 2018). A recent study by Chen et al (Cancer Cell, 2021) has further showed that depletion of type I collagen in CAFs aggravates tumor progression in murine models.

The paper would therefore be strengthened by including data on an experimental model without co-injection of skin-fibroblasts. The data from the orthotopic breast cancer model where 4T1 cells and skin fibroblasts were injected to the breast fat pad are impressive, but it would be interesting to see if similar results could be obtained in a model where tissue resident fibroblasts would form a desmoplastic stroma, for instance in a KPC model.

Another aspect to consider when studying the VDR-signalling is the lack of conservation in vitamin D3 response elements in non-primates. Many of the genes that are activated in human cells after VDR-signalling are not induced in murine cells (Dimitrov et al, J Steroid Biochem Mol Biol 2016). For instance, vitamin D3 can induce production of the anti-microbial peptide cathelicidin in humans but not in mice, which when cleaved to the active form LL-37 is involved in host defence and

immunomodulation. High levels of LL-37 have been associated with cancer stem cell growth and survival and pro-angiogenic effects in human pancreatic cancer (Sainz et al Gut 2015). Tumor tissues from the mice were used to examine localization of Cal/ICG ex vivo. Here it would be more informative to use human tumor tissues since this will determine both if the Cal/ICG@MP is superior in delivering the substance to the tumor cells in a human model and it would also represent a more realistic and "true to life" stromal microenvironment.

Fig 3. Were the T cells stimulated in the presence of DCs? Or is the response only directed to tumor-derived antigens presented by DCs? It seems like a very robust response considering that the T cells were derived from tumor naïve animals.

Fig 4. It is stated that free Cal and Cal/ICG significantly decreased the expression of α -SMA. This is only showed by a representative (?) western blot. Was this experiment repeated more than once? There are studies suggesting that the protein expression of α -SMA is decreased in human CAFs by calcipotriol (Gorchs et al, Scientific reports, 2020).

Reviewer #3:

Remarks to the Author:

The manuscript from Li et al., investigates the effect of deactivating cancer-associated fibroblasts (CAFs) for improved photothermal therapy (PTT) efficacy both in vitro and in vivo. The authors constructed a tumor cell-derived microparticles co-delivering calcipotriol and Indocyanine green (Cal/ICG@MPs). Cal/ICG@MPs efficiently deactivate intratumoral CAFs to a quiescent state and remodel tumor extracellular matrix (ECM), resulting in the enhanced tumor accumulation and penetration of ICG. In addition, Cal/ICG@MPs exhibits strong systemic antitumor immune response and induces a long-term immunological memory function, generating excellent antitumor effects against not only primary large tumors, but also untreated distant tumors and even metastatic tumors via abscopal effects. The data was somewhat efficacious to support the conclusions and claims. The phenotypic aspects of this investigation however are not novel with it previously established that photothermal can deplete cancer-associated fibroblasts to normalizes tumor stiffness (Alba Nicolás-Boluda, 2020, ACS NANO 14, 5, 5738–5753) as well as combination therapy of CAF depletion and PTT (Jingchao Li, ACS Nano 2018, 12, 8, 8520–8530). There are a number of significant concerns, and the most substantial are laid out below, and can not be published at present.

Major Concerns:

1. The advancement of this system compared to previous research has not emphasized within the introduction.
2. It has been previously reported that photothermal can deplete cancer-associated fibroblasts to normalizes tumor stiffness (Alba Nicolás-Boluda, 2020, ACS NANO 14, 5, 5738–5753). So studying the influence of PTT on the CAF depletion is suggested.
3. How to remove the unloaded Cal and ICG during the preparation of Cal/ICG@MPs?
4. The author used MPs from H22 cells, so stronger cytotoxicity and tumor inhibition effect was detected in the ICG@MPs- and Cal/ICG@MPs group compared with free ICG- or Cal/ICG-treated group. But why similar results were detected in murine 4T1 breast cancer cells? Why author used 4T1 tumor model? Does the MPs can exhibit non-homologous targeting effect?

Minor concerns:

1. The semi-quantitation of WB results should be added.
2. The relationship of ICD and AICD should be discussed more specific.

Reviewer #4:

Remarks to the Author:

The work from Li and Yong et al aims at improving the efficacy of photothermal therapy (PTT) for cancer treatment by utilising microparticles (MPs) co-delivering a cancer-associated fibroblast

(CAF)-targeting agent (i.e. calcipotriol) for stroma remodelling, in addition to a NIR fluorescence dye (i.e. Indocyanine green (ICG)) for PTT. The authors show that these new tumor-derived microparticles lead to (1) more effective accumulation in cancer cells in vitro and in tumors in vivo, (2) reduction in myofibroblasts (see comments later on), (3) increased maturation of dendritic cells and activation of CD8+ T cells in vitro and in vivo and (4) impairment of tumour growth and metastasis formation in vivo.

The work is of interest and potentially translatable to various stroma-rich cancer types. The majority of conclusions is supported by a significant amount of data. The two main concerns (see for details below) are linked to (1) translational and human relevance and (2) the need to provide additional evidence in support of the "CAF deactivation" claim. Addressing these concerns and/or rephrasing certain passages, should make this manuscript suitable for publication in Nature Communications.

Main points

1) The claim that the increased efficacy of the Cal/ICG MPs is due to deactivation and not depletion of CAFs needs further evidence otherwise the title and text should be rephrased.

a. More evidence is needed to support the claim that deactivation/depletion of CAFs is what mediates the increased efficacy of Cal/ICG MPs compared to other groups. A way to test this would be to evaluate whether in non-stroma rich tumours the differences in efficacy between treatments would be more attenuated compared to "stroma-rich" tumors. Moreover, the orthotopic mammary models, and most of the subcutaneous models, have not been well characterised in terms of ECM and fibroblast presence (also see below).

b. I believe the authors only utilise one line of skin fibroblasts for the in vitro and co-transplantation experiments. The authors should repeat at least some of these experiments with a second fibroblast line. Ideally, with hepatic stellate cells when they use the murine hepatocarcinoma cell line H22.

c. The data regarding the reduction in collagen deposition upon delivery of Cal/ICG MPs need improvement. (1) The Masson trichrome's stains shown (e.g. Fig 4d) are far from being clear – new images and quantification should be provided; (2) Sirius red quantification (e.g. 4b) should be accompanied by representative images to show the quality of the stain.

d. Moreover, while it is true that calcipotriol has been shown to bring back CAFs to quiescence, additional evidence that this is the mechanism in the models used is required. Reduction of collagen or aSMA levels are not definitive evidence that CAFs become quiescent rather than be depleted in vivo. Of note, aSMA is also a pericyte marker so single stain is not sufficient to address this point. Moreover, Fig 4C supports the depletion of CAFs by showing a reduction in percentage of fibroblasts. To start addressing this, the authors could stain tissues for a pan-CAF marker, such as PDPN, an epithelial marker (e.g. ECAD) and aSMA (marker of activated CAFs) and evaluate whether PDPN+ECAD-SMA- and SMA+PDPN+ECAD- cell numbers change. Moreover, collagen analysis should be included in all the in vivo experiments shown (e.g Fig 6,7,8,9).

e. In Fig 4I, it is not very evident that Cal/ICG MPs are significantly better than ICG MPs. Quantification of all replicates should be included to demonstrate this point convincingly.

f. The authors also show similar immune changes in the spleen and lymph nodes compared to tumours with better efficacy of Cal/ICG MPs compared to other groups. Why is the presence of Cal in the MPs be more effective than ICG alone? The authors should provide additional immunochemical and histological evidence if they believe it is also associated to CAFs/matrix, which should not be present at these sites.

g. Figure 6f should be quantified. From the images, it appears that aSMA is reduce also by ICG MPs.

2) Can the authors provide any data that this is relevant to the human disease? Clearly the immune aspect of the manuscript cannot be validated in vivo and would be challenging to address in vitro. The authors could, however, repeat some of the in vitro and/or in vivo experiments with at least one human cancer cell line (and fibroblast line) to at least show increased accumulation in the presence of Cal/ICG MPs compared to other groups etc.

3) The majority of flow data in respect to immune cell populations show specific markers within that population. Can the authors also include the total number (i.e. from live singlets) of the main populations central to their story (i.e. T cells, CD8+ T cells, Tregs, DCs)? This is particularly

relevant for T cells and CD8+ T cells if the authors want to claim increased T cell infiltration, not only activation, in tumours and in vitro. Especially, since it appears that Cal/ICG MPs is better than ICG MPs in this regard, but not compared to other groups (e.g. Fig j-k).

Minor points

- 1) Could the authors please provide the data described on lines 123-126? I believe it is not included in the manuscript, although mentioned there.
- 2) Can the authors please clarify the rationale in the methods for the different time point chosen for the experiments in vitro (e.g. 24h for DCs, 5 days DCs + T cells, ...).
- 3) Can the authors include in their analysis of T cells also exhaustion markers to evaluate whether they decrease in parallel to the increase in activation markers?
- 4) The authors should not utilise the term "CAFs" when talking about fibroblasts that have not been exposed to tumour cues, and instead should use the term "myofibroblasts".
- 5) In addition to CD45 and CD31, an epithelial marker (e.g. EpCAM and/or ECAD) should have also been included in the analysis shown in 4C as PDPN and PDGFRA can in some cases be expressed by cancer cells.
- 6) Could the authors please clarify whether Fig.6i is significant between Cal/ICG MPs and other groups?
- 7) Can the authors please provide zoomed in pictures for the tumours in Supp Fig 20 and clarify whether there is increased necrosis upon treatment with Cal/ICG MPs? The authors should clarify what they mean saying that the treatment is effective based on those images.
- 8) In the chemotaxis experiment (supp fig 14), can the authors also provide the flow analysis for the activation markers of the CD8+ T cells?
- 9) The change in solid stress of tumours was only shown for the 4T1 models, can the authors provide this for the H22 models or state in the main text that this has not been assessed in that context?
- 10) Do Cal/ICG MPs affect cancer cell and fibroblast' proliferation in vivo more than other groups?

RESPONSES TO REVIEWERS

We would like to express our sincere thanks to all four reviewers for their critical and constructive comments. We have performed substantial additional experiments to address their concerns. We respond point-by-point to each of their comments and criticisms. We feel that their comments have helped us on significantly improving and strengthening the manuscript and clarifying some issues. We hope that the revision has addressed their major concerns.

Response to Reviewer 1

In this manuscript, the authors designed a cancer cell-derived microparticles co-delivering calcipotriol and Indocyanine green (Cal/ICG@MPs) to deactivate CAFs for improved PTT efficacy. With the enhanced accumulation and distribution of the ICG in the tumor, a strong PTT efficacy and activation of CD8⁺ T cell-mediated antitumor immunity were observed. In addition, the authors also showed the long-term antitumor immune memory to inhibit tumor recurrence and metastasis via Cal/ICG@MPs. Overall, it was well organized and written. My concerns are listed below.

1. It is challenging to maintain the stability of the cell-derived microparticles in the aqueous solution. The authors demonstrated no significant size and zeta potential change of Cal/ICG@MPs over 7 days. Please give the explanation why these particles were stable.

Response: We thank the reviewer's critical questions. We agreed with the reviewer that microparticles (MPs) are heterogeneous populations of cell-derived vesicles. In this work, we just evaluated the size and zeta potential of Cal/ICG@MPs for 7 days. It seemed that Cal/ICG@MPs were stable during 7 days. Similar results were shown in *Nature Communications* 2019,10: 3288; *Nature Biomedical Engineering* 2019, 3: 729; *Nature Communications* 2021, 12: 440, et al. Investigating the stability of Cal/ICG@MPs for a longer period is needed in future work. Meanwhile, the development of proper storage techniques, such as lyophilization, to remain the activity of Cal/ICG@MPs is also required for future clinical translation.

2. The results supported Cal/ICG@MPs as a promising drug to improve PTT efficacy in cancer treatment. Did the authors try the ICG@MPs and Cal combination formulation? Whether it is better to treat the mice with Cal and followed by ICG@MPs?

Response: We thank the reviewer's constructive suggestion. Accordingly, we constructed the stroma-rich H22 tumor-bearing mice by co-injection of H22 cells and hepatic stellate cells (HSCs), and then determined the in vivo anticancer activity by intravenous injection of PBS, ICG, Cal/ICG, ICG@MPs, Cal@MPs, ICG@MPs+Cal (ICG@MPs and Cal combination formulation) or

Cal/ICG@MPs twice every two days, followed with or without 808 nm laser irradiation. Consistently, Cal/ICG@MPs exhibited the strongest anticancer activity upon 808 nm laser irradiation, further confirming that co-delivery of Cal and ICG by MPs to tumor tissues generated the strongest anticancer activity upon laser irradiation.

We added these data in Supplementary Figure 37 in the revised manuscript.

3. Due to the different functions and targeted cells of the Cal and ICG, the rationale of combining the Cal and ICG in the microparticles should be discussed.

Response: We thank the reviewer's critical comments. We agreed with the reviewer that Cal should be internalized into CAFs to deactivate CAFs, while ICG should be internalized into tumor cells to trigger PTT-induced cytotoxicity. In our work, we used MPs derived from tumor cells to co-deliver Cal and ICG for efficient PTT. The reasons were that: (1) MPs can efficiently deliver both Cal and ICG to tumor tissues, enhancing the tumor accumulation of Cal and ICG to exert stronger functions. (2) Although MPs co-delivering Cal and ICG can target to tumor tissues, more MPs were accumulated in tumor cells compared with CAFs in tumor tissues, generating strong photothermal effects against tumor cells. (3) Although MPs co-delivered ICG and Cal to CAFs, CAFs were resistant to ICG-triggered PTT compared with tumor cells. In contrast, Cal delivered by MPs efficiently deactivated CAFs to remodel tumor ECM and ameliorate CAF-induced antigen-mediated activation-induced cell death (AICD) of tumor-specific CD8⁺ T cells in response to Cal/ICG@MPs-triggered PTT. Taken together, although Cal and ICG had different functions for different targeted cells, Cal/ICG@MPs fully acted on the photothermal and CAF deactivation effects to achieve desirable anticancer activity, which was confirmed by the stronger anticancer activity of Cal/ICG@MPs with 808 nm laser irradiation-treated group than the combination of ICG@MPs and Cal-treated group.

We added the related information in the Discussion section in the revised manuscript.

4. Although the ICG and Cal were loaded in the microparticles, whether the ratio of ICG and Cal in the manuscript was the best one. In addition, the drug loading and release mechanism should be studied.

Response: We thank the reviewer's critical questions. In fact, we optimized the drug loading condition. First, we incubated tumor cells with different concentrations of ICG and then collected ICG@MPs for determination of ICG loading efficiency. Our results showed that when the feeding concentration of ICG was 100 $\mu\text{g mL}^{-1}$, the ICG loading efficiency almost reached the plateau, thus we set the feeding concentration of ICG at 100 $\mu\text{g mL}^{-1}$. Furthermore, when the tumor cells were incubated with ICG at 100 $\mu\text{g mL}^{-1}$ and different

concentrations of Cal, we collected Cal/ICG@MPs and determined the loading efficiency of Cal and ICG. The results showed that loading Cal did not significantly affect the ICG loading efficiency. Although the Cal loading efficiency was higher as feeding more Cal, we set the feeding concentration of Cal at $4 \mu\text{g mL}^{-1}$ considering that the present Cal loading efficiency met the requirements for in vitro and in vivo experiments according to the reported literatures.

About the drug loading mechanism, we developed Cal/ICG@MPs by incubating tumor cells with Cal and ICG and then collecting MPs by ultracentrifugation in this work. Cal/ICG@MPs were constructed by virtue of the internalization of Cal and ICG by tumor cells and then budding from cell plasma membranes. Similar drug loading approaches for MPs were found in *Nature Communications* 2012, 3: 1282; *Nature Biomedical Engineering* 2019, 3: 729–740; *Nature Communications* 2021, 12: 440, et al.

To determine the release mechanism of Cal/ICG@MPs, we measured the drug release of Cal/ICG@MPs in PBS at different pH values. Our results showed that more Cal was released at pH 4.5 (lysosomal acidic pH) compared with pH 6.5 and 7.4, exhibiting pH-responsive sustained drug-release profiles. These results suggested that Cal/ICG@MPs might be internalized into cells and release drugs in lysosomes to exert the biological function.

We added the drug loading efficiency at different feeding concentrations in Supplementary Table 1 and 2, and drug release profiles at different pH values in Supplementary Figure 1 in the revised manuscript.

5. Stroma-rich H22 tumor-bearing mice were utilized for the in vivo study. Whether Cal/ICG@MPs could accumulate in the CAFs? What's the PTT effect to this type of cells?

Response: We thank the reviewer's constructive questions. Accordingly, we constructed stroma-rich GFP-expressing H22 tumor-bearing mice by co-injection of CAFs and GFP-stably expressing H22 cells. The mice were intravenously injected with Cal/ICG@MPs and at 48 h after injection, the overlay of Cal/ICG@MPs with GFP⁺ H22 cells or CAFs (labeling with α -SMA antibody) was determined by confocal microscopy. The results showed that Cal/ICG@MPs efficiently colocalized with tumor cells and CAFs, suggesting that Cal/ICG@MPs can be accumulated in tumor cells and CAFs. However, more Cal/ICG@MPs were accumulated in tumor cells compared with those in CAFs, which might provide the possibility of stronger PTT against tumor cells, but not CAFs to deplete them.

To determine the cytotoxicity of Cal/ICG@MPs against CAFs upon 808 nm laser irradiation, CAFs were treated with ICG, Cal/ICG, ICG@MPs, Cal@MPs or Cal/ICG@MPs for 4 h, followed with or without 808 nm laser irradiation. In contrast to tumor cells (Figure 3a in the revised manuscript), no significant cytotoxicity was detected in ICG-, Cal/ICG-, ICG@MPs- and Cal/ICG@MPs-

treated groups upon laser irradiation, suggesting that CAFs were resistant to PTT compared with tumor cells.

We added these data in Supplementary Figure 20 in the revised manuscript.

Response to Reviewer 2

Li et al have investigated means to increase the effectiveness of photothermal therapy by delivering the substance indocyanine green to tumor cells by tumor derived microparticles conjugated with calcipotriol. They show that these Cal/ICG@MPs effectively target the tumor, remodel the tumor microenvironment by deactivating CAFs, increase tumor-specific immunity and inhibit tumor growth. This is very interesting concept and the results look promising but there are some concerns that need to be addressed.

(1) CAFs were generated from healthy murine skin explants and cultured in TGF-beta. This seems a bit simplistic since there are many other factors in the tumor microenvironment that can drive the differentiation of CAFs. Fibroblasts isolated from skin may also inherently be very different compared to fibroblasts isolated from other tissues. For instance, the majority of fibroblasts in the liver are stellate cells which can react differently compared to skin-derived fibroblasts in the presence of tumor cells. It has also recently been shown that there are several subtypes of CAFs with diverse functions and the subgroup expressing high levels of α -SMA (myCAF) appears to protect from tumor invasion rather promote malignancy (Öhlund et al JEM, 2017 Biffi et al Cancer Discov 2019, Kieffer et al Cancer Discov 2020, Costa et al Cancer Cell 2018). A recent study by Chen et al (Cancer Cell, 2021) has further showed that depletion of type I collagen in CAFs aggravates tumor progression in murine models.

The paper would therefore be strengthened by including data on an experimental model without co-injection of skin-fibroblasts. The data from the orthotopic breast cancer model where 4T1 cells and skin fibroblasts were injected to the breast fat pad are impressive, but it would be interesting to see if similar results could be obtained in a model where tissue resident fibroblasts would form a desmoplastic stroma, for instance in a KPC model.

Response: We thank the reviewer's constructive suggestion. Accordingly, we constructed the stroma-poor H22 tumor-bearing mice without co-injection of skin fibroblasts, and then determined the anticancer activity after intravenous injection of PBS, ICG, Cal/ICG, ICG@MPs or Cal/ICG@MPs, followed with 808 nm laser irradiation. Different from the stroma-rich H22 tumor-bearing model showing that Cal/ICG@MPs exhibited stronger anticancer activity than ICG@MPs upon 808 nm laser irradiation, Cal/ICG@MPs and ICG@MPs exhibited similar anticancer activity upon 808 nm laser irradiation in stroma-poor H22 tumor-bearing mice, confirming that co-delivery of Cal to deactivate CAFs efficiently contributed to the anticancer activity of ICG@MPs upon laser irradiation.

To further verify CAF deactivation by Cal/ICG@MPs, azoxymethane (AOM)/dextran sodium sulfate (DSS)-induced spontaneous colitis-associated cancer (CAC) mouse model was constructed and then intravenously injected with Cal/ICG, ICG@MPs, Cal@MPs or Cal/ICG@MPs every two days for three times. Consistently, significantly lower expression of α -SMA by immunofluorescence analysis and decreased deposition of collagen fibers by Masson's trichrome staining were detected in Cal@MPs- and Cal/ICG@MPs-treated groups, confirming that Cal/ICG@MPs efficiently deactivated CAFs.

We added these results in Supplementary Figure 38 and 22 in the revised manuscript.

(2) Another aspect to consider when studying the VDR-signalling is the lack of conservation in vitamin D3 response elements in non-primates. Many of the genes that are activated in human cells after VDR-signalling are not induced in murine cells (Dimitrov et al, J Steroid Biochem Mol Biol 2016). For instance, vitamin D3 can induce production of the anti-microbial peptide cathelicidin in humans but not in mice, which when cleaved to the active form LL-37 is involved in host defense and immunomodulation. High levels of LL-37 have been associated with cancer stem cell growth and survival and pro-angiogenic effects in human pancreatic cancer (Sainz et al Gut 2015). Tumor tissues from the mice were used to examine localization of Cal/ICG ex vivo. Here it would be more informative to use human tumor tissues since this will determine both if the Cal/ICG@MP is superior in delivering the substance to the tumor cells in a human model and it would also represent a more realistic and "true to life" stromal microenvironment.

Response: We thank the reviewer's constructive suggestion. Accordingly, organotypic tumor slices from liver cancer patients were treated with PBS, Cal/ICG, ICG@MPs, Cal@MPs or Cal/ICG@MPs for 48 h. Consistently, Cal@MPs and Cal/ICG@MPs significantly decreased the expression of α -SMA, fibronectin and collagen-I by immunofluorescence analysis, and reduced the deposition of collagen fibers by Masson's trichrome staining, suggesting that Cal/ICG@MPs efficiently remodeled tumor ECM through CAF deactivation in human tumor tissues.

Furthermore, organotypic tumor slices from liver cancer patients were treated with ICG, Cal/ICG, ICG@MPs or Cal/ICG@MPs for 48 h and then determined ICG distribution in tumor tissues. Consistently, the strongest ICG fluorescence was detected in Cal/ICG@MPs-treated group. Meanwhile, more Cal/ICG@MPs were spread inside the tumor tissues compared with other groups, suggesting that Cal/ICG@MPs-induced tumor ECM remodeling might contribute to their enhanced tumor accumulation and penetration in human tumor tissues.

We added these data in Supplementary Figure 16 and 25 in the revised manuscript.

(3) Fig 3. Were the T cells stimulated in the presence of DCs? Or is the response only directed to tumor-derived antigens presented by DCs? It seems like a very robust response considering that the T cells were derived from tumor naïve animals.

Response: We thank the reviewer's critical comments. To determine CD8⁺ T cell activation, we first treated H22 cells with PBS, ICG, Cal/ICG, ICG@MPs, Cal@MPs or Cal/ICG@MPs for 4 h, followed with or without 808 nm laser irradiation. Furthermore, these treated H22 cell supernatants containing tumor antigens were harvested and added to immature bone marrow-derived DCs (BMDCs) for 24 h to stimulate DC maturation. These matured BMDCs were then incubated with CD3⁺ T cells (isolated from spleens and expanded in medium containing 20 ng mL⁻¹ IL-2 for 5 days) at a BMDC/T cell ratio of 1:10 for 5 days. The results showed that the higher percentages of CD8⁺IFN- γ ⁺ T, CD8⁺IL-2⁺ T, CD8⁺TNF- α ⁺ T and CD8⁺Granzyme B⁺ T cells were detected in ICG@MPs- and Cal/ICG@MPs-treated groups upon 808 nm laser irradiation, suggesting that DCs matured by Cal/ICG@MPs with 808 nm laser irradiation-treated H22 cells exhibited the strong capacity to activate CD8⁺ T cells. The similar CD8⁺ T cell activation by the matured DCs was detected in *Nature Communications* 2020, 11: 1790; *Advanced Materials* 2021, 33: e2006003, et al.

(4) Fig 4. It is stated that free Cal and Cal/ICG significantly decreased the expression of α -SMA. This is only showed by a representative western blot. Was this experiment repeated more than once? There are studies suggesting that the protein expression of α -SMA is decreased in human CAFs by calcipotriol (Gorchs et al, Scientific reports, 2020).

Response: We thank the reviewer's constructive suggestion. We performed three independent western blot experiments for Fig. 4a. Our results showed that α -SMA expression was significantly decreased in CAFs after treatment with Cal, Cal/ICG, Cal@MPs or Cal/ICG@MPs. The semi-quantitation results of three independent experiments were shown in Supplementary Figure 11 in the revised manuscript. The similar phenomena were shown in *Cell*, 2014, 159: 80-93, in which Cal significantly decreased the expression of *ACTA2* (α -SMA) in pancreatic stellate cells isolated from patients with pancreatic ductal adenocarcinoma.

Response to Reviewer 3

The manuscript from Li et al., investigates the effect of deactivating cancer-associated fibroblasts (CAFs) for improved photothermal therapy (PTT) efficacy both in vitro and in vivo. The authors constructed a tumor cell-derived microparticles co-delivering calcipotriol and Indocyanine green

(Cal/ICG@MPs). Cal/ICG@MPs efficiently deactivate intratumoral CAFs to a quiescent state and remodel tumor extracellular matrix (ECM), resulting in the enhanced tumor accumulation and penetration of ICG. In addition, Cal/ICG@MPs exhibits strong systemic antitumor immune response and induces a long-term immunological memory function, generating excellent antitumor effects against not only primary large tumors, but also untreated distant tumors and even metastatic tumors via abscopal effects. The data was somewhat efficacious to support the conclusions and claims. The phenotypic aspects of this investigation however are not novel with it previously established that photothermal can deplete cancer-associated fibroblasts to normalizes tumor stiffness (Alba Nicolás-Boluda, 2020, ACS NANO 14, 5, 5738–5753) as well as combination therapy of CAF depletion and PTT (Jingchao Li, ACS Nano 2018, 12, 8, 8520–8530). There are a number of significant concerns, and the most substantial are laid out below, and can not be published at present.

Major Concerns:

1. The advancement of this system compared to previous research has not emphasized within the introduction.

Response: We thank the reviewer's constructive suggestion. Accordingly, we supplemented the previous research in the Introduction section about depleting CAFs using PTT (*ACS Nano* 2020, 14: 5738) and combination therapy of CAF depletion and PTT (*ACS Nano* 2018, 12: 8520) for cancer treatment, in which these constructed photothermal agents were specifically targeted to CAFs compared with tumor cells. However, recent studies showed that CAF depletion may run the risk of ablating vital stromal components that are required for tissue homeostasis, further exacerbating the tumor growth (*Cancer Cell* 2014, 25: 719; *Nature Reviews Drug Discovery* 2019, 18: 99). Reprogramming the tumor stroma via CAF deactivation, rather than targeted ablation of the CAFs per se, may be the preferable therapeutic paradigm for cancer therapy (*Nature Reviews Drug Discovery* 2019, 18: 99). Thus, we developed the tumor cell-derived microparticles co-delivering ICG and calcipotriol (denoted as Cal/ICG@MPs) for cancer therapy. Cal/ICG@MPs specifically targeted to tumor tissues, especially tumor cells rather than CAFs (although Cal/ICG@MPs can be internalized by CAFs) for efficient PTT against tumor cells, while calcipotriol efficiently deactivated, not depleted CAFs for decreased ECM remodeling to enhance ICG tumor accumulation and penetration and CD8⁺ T cell intratumoral distribution, as well as ameliorated CAF-induced antigen-mediated AICD of CD8⁺ T cells in response to PTT. We think that this design concept of Cal/ICG@MPs is the main innovation of our work.

2. It has been previously reported that photothermal can deplete cancer-associated fibroblasts to normalizes tumor stiffness (Alba Nicolás-Boluda, 2020,

ACS NANO 14, 5, 5738–5753). So studying the influence of PTT on the CAF depletion is suggested.

Response: We thank the reviewer's constructive suggestion. We constructed stroma-rich H22 tumor-bearing mice and then intravenously injected with PBS, Cal/ICG, ICG@MPs, Cal@MPs or Cal/ICG@MPs twice every two days, followed with or without 808 nm laser irradiation. Our results showed that although Cal@MPs- and Cal/ICG@MPs significantly decreased the numbers of CAFs (PDPN⁺CD140 α ⁺CD45⁻CD31⁻EpCAM⁻ cells or PDPN⁺ α -SMA⁺CD45⁻CD31⁻EpCAM⁻ cells), and reduced α -SMA, fibronectin, collagen I expression and collagen fiber deposition in tumor tissues, 808 nm laser irradiation did not markedly promote Cal/ICG@MPs-induced decrease in the numbers of CAFs and the tumor stromal reprogramming. These results suggested that Cal/ICG@MPs efficiently deactivated CAFs, while Cal/ICG@MPs-triggered PTT did not further deplete CAFs. In addition, in vitro MTT assay also showed that TGF- β -activated skin fibroblasts were resistant to Cal/ICG@MPs-triggered PTT.

We supplemented these data in the Figure 4c, 4d and Supplementary Figure 18a, 20, 21 in the revised manuscript.

3. How to remove the unloaded Cal and ICG during the preparation of Cal/ICG@MPs?

Response: We thank the reviewer's critical questions. In this work, Cal/ICG@MPs were prepared by incubating H22 or 4T1 cells with 100 $\mu\text{g mL}^{-1}$ ICG and 4 $\mu\text{g mL}^{-1}$ Cal for 12 h, followed by centrifuging the supernatants at 600g for 10 min and 18,000g for 2 min to remove the debris, and then centrifuging the supernatants at 18,000g for 30 min to harvest Cal/ICG@MPs. This ultracentrifugation process removed most of the unloaded Cal and ICG. To further remove the left unloaded Cal and ICG, the harvested Cal/ICG@MPs were washed with PBS and ultracentrifuged for three times. No significant fluorescence signal of ICG and UV absorbance of Cal were observed in the centrifuged washing solution after three times washing, suggesting that the unloaded Cal and ICG were completely removed. The detailed information on the preparation and purification of Cal/ICG@MPs was shown in the Methods section in the revised manuscript.

4. The author used MPs from H22 cells, so stronger cytotoxicity and tumor inhibition effect was detected in the ICG@MPs- and Cal/ICG@MPs group compared with free ICG- or Cal/ICG-treated group. But why similar results were detected in murine 4T1 breast cancer cells? Why author used 4T1 tumor model? Does the MPs can exhibit non-homologous targeting effect?

Response: We are sorry that we did not make it clear in the manuscript. To clarify the functional universality of Cal/ICG@MPs, we used different tumor

cells and tumor-bearing mice, such as H22 cells, 4T1 cells, H22 tumor-bearing mice, and 4T1 tumor-bearing mice. We did not evaluate whether MPs exhibit non-homologous targeting effect in this manuscript. We used Cal/ICG@MPs derived from H22 cells when we performed the cytotoxicity against H22 cells and the anticancer activity in H22 tumor-bearing mice. Furthermore, we used Cal/ICG@MPs derived from 4T1 cells to evaluate the cytotoxicity against 4T1 cells and the anticancer activity in 4T1 tumor-bearing mice. We supplemented the detailed information in the revised manuscript.

Minor concerns:

1. The semi-quantitation of WB results should be added.

Response: We thank the reviewer's constructive question. We performed three independent experiments for each western blot data. According to the reviewer's suggestion, the semi-quantitation analysis of western blots was carried out. We added these semi-quantitation results in Supplementary Figure 10 and 11 in the revised manuscript.

2. The relationship of ICD and AICD should be discussed more specific.

Response: We thank the reviewer's constructive suggestion. Successful CTL-mediated tumor rejection requires the recruitment, infiltration and expansion of tumor antigen-specific CTLs in tumor tissues. CAFs have been shown to abrogate CD8⁺ T cell function through restricting CTL recruitment to tumor tissues, and producing aberrant ECM deposition to function as a physical barrier against tumor infiltration of CTLs. Meanwhile, after the recruitment of CD8⁺ T cell to tumor tissues, CAFs in the tumor stroma, promote cancer immune evasion by inducing AICD of the tumor-specific CD8⁺ T cells. Mechanistically, CAFs can process and cross-present tumor antigens via the class I major histocompatibility complex to provide repeated T cell receptor (TCR) ligation on the tumor-specific CTLs. Additionally, CAFs express large amounts of FAS ligand (FASL), which induces apoptosis of FAS-expressing CD8⁺ T cells, and CAFs also express programmed cell death 1 ligand 2 (PD-L2), which induces T cell anergy by interacting with the immune checkpoint molecule PD-1.

In this work, Cal/ICG@MPs-triggered PTT efficiently induced ICD effects of tumor cells, resulting in DC maturation and CD8⁺ T cell activation, which mainly occurs in LNs. Cal/ICG@MPs-induced CAF deactivation allowed the recruitment of the activated CD8⁺ T cells to tumor tissues, and meanwhile remodeled tumor ECM to improve CD8⁺ T cell infiltration to deep tumor parenchyma. Importantly, Cal/ICG@MPs-caused CAF deactivation significantly decreased CAF-induced AICD of these CD8⁺ T cells when CAFs processed and cross-presented tumor antigens derived from the Cal/ICG@MPs-triggered PTT-induced ICD of tumor cells, as evidenced by the decreased expression of

MHC-1, PD-L2 and FasL in CAFs after treatment with Cal/ICG@MPs upon 808 nm laser irradiation and the correspondingly enhanced CD8⁺ T cell proliferation and decreased CD8⁺ T cell apoptosis. Thus, Cal/ICG@MPs significantly increased CD8⁺ T cell number in tumor tissues upon nm laser irradiation compared with ICG@MPs with 808 nm laser irradiation-treated group, generating long-term antitumor immune memory to inhibit tumor recurrence and metastasis.

We supplemented these details in the Discussion section in the revised manuscript.

Response to Reviewer 4

The work from Li and Yong et al aims at improving the efficacy of photothermal therapy (PTT) for cancer treatment by utilising microparticles (MPs) co-delivering a cancer-associated fibroblast (CAF)-targeting agent (i.e. calcipotriol) for stroma remodelling, in addition to a NIR fluorescence dye (i.e. Indocyanine green (ICG)) for PTT. The authors show that these new tumor-derived microparticles lead to (1) more effective accumulation in cancer cells in vitro and in tumors in vivo, (2) reduction in myofibroblasts (see comments later on), (3) increased maturation of dendritic cells and activation of CD8⁺ T cells in vitro and in vivo and (4) impairment of tumour growth and metastasis formation in vivo.

The work is of interest and potentially translatable to various stroma-rich cancer types. The majority of conclusions is supported by a significant amount of data. The two main concerns (see for details below) are linked to (1) translational and human relevance and (2) the need to provide additional evidence in support of the “CAF deactivation” claim. Addressing these concerns and/or rephrasing certain passages, should make this manuscript suitable for publication in Nature Communications.

Main points

(1) The claim that the increased efficacy of the Cal/ICG MPs is due to deactivation and not depletion of CAFs needs further evidence otherwise the title and text should be rephrased.

a. More evidence is needed to support the claim that deactivation/depletion of CAFs is what mediates the increased efficacy of Cal/ICG MPs compared to other groups. A way to test this would be to evaluate whether in non-stroma rich tumours the differences in efficacy between treatments would be more attenuated compared to “stroma-rich” tumors.

Response: We thank the reviewer’s constructive question. Accordingly, we constructed the stroma-poor H22 tumor-bearing mice without co-injection of CAFs, and then determined the anticancer activity after intravenous injection of PBS, ICG, Cal/ICG, ICG@MPs or Cal/ICG@MPs twice every two days, followed with 808 nm laser irradiation. Different from the stroma-rich H22 tumor-bearing model showing that Cal/ICG@MPs exhibited stronger anticancer

activity than ICG@MPs upon 808 nm laser irradiation, Cal/ICG@MPs and ICG@MPs exhibited similar anticancer activity upon laser irradiation in stroma-poor H22 tumor-bearing mice, confirming that CAFs deactivated by Cal efficiently contributed to the anticancer activity of ICG@MPs upon laser irradiation.

We added these results in Supplementary Figure 38 in the revised manuscript.

Moreover, the orthotopic mammary models, and most of the subcutaneous models, have not been well characterized in terms of ECM and fibroblast presence (also see below).

b. I believe the authors only utilise one line of skin fibroblasts for the in vitro and co-transplantation experiments. The authors should repeat at least some of these experiments with a second fibroblast line. Ideally, with hepatic stellate cells when they use the murine hepatocarcinoma cell line H22.

Response: We thank the reviewer's constructive suggestion. Accordingly, we constructed stroma-rich H22 tumor-bearing mice by co-injection of H22 cells and hepatic stellate cells (HSCs), and then intravenously injected with PBS, ICG, Cal/ICG, ICG@MPs or Cal/ICG@MPs twice every two days, followed with 808 nm laser irradiation (1.5 W cm^{-2} , 10 min) at 2 h after the last injection. Consistently, Cal/ICG@MPs-treated group exhibited maximal temperature in tumor tissues upon 808 nm laser irradiation for 10 min. Furthermore, the anticancer activity was determined in these stroma-rich H22 tumor-bearing mice after intravenous injection of PBS, ICG, Cal/ICG, ICG@MPs, Cal@MPs or Cal/ICG@MPs twice every two days, followed with or without 808 nm laser irradiation at the tumor sites for 10 min. Consistently, the strongest tumor inhibition activity was detected in Cal/ICG@MPs-treated group upon 808 nm laser irradiation. These results confirmed that Cal/ICG@MPs exhibit the excellent photothermal effects and anticancer activity in stroma-rich H22 tumor-bearing mice constructed by co-injection of H22 cells and TGF- β -treated skin fibroblasts or HSCs.

We added these results in Supplementary Figure 26 and 37 in the revised manuscript.

c. The data regarding the reduction in collagen deposition upon delivery of Cal/ICG MPs need improvement. (1) The Masson trichrome's stains shown (e.g. Fig 4d) are far from being clear – new images and quantification should be provided; (2) Sirius red quantification (e.g. 4b) should be accompanied by representative images to show the quality of the stain.

Response: We thank the reviewer's constructive suggestion. Accordingly, we used the new images of Masson trichrome's staining with a larger magnification and supplemented their quantification. Meanwhile, we added the representative images of Sirius red staining.

We added these data in Figure 4d and Supplementary Figure 21, 14 in the revised manuscript.

d. Moreover, while it is true that calcipotriol has been shown to bring back CAFs to quiescence, additional evidence that this is the mechanism in the models used is required. Reduction of collagen or α SMA levels are not definitive evidence that CAFs become quiescent rather than be depleted in vivo. Of note, α SMA is also a pericyte marker so single stain is not sufficient to address this point. Moreover, Fig 4C supports the depletion of CAFs by showing a reduction in percentage of fibroblasts. To start addressing this, the authors could stain tissues for a pan-CAF marker, such as PDPN, an epithelial marker (e.g. ECAD) and α SMA (marker of activated CAFs) and evaluate whether PDPN+ECAD- α SMA- and SMA+PDPN+ECAD- cell numbers change.

Moreover, collagen analysis should be included in all the in vivo experiments shown (e.g Fig 6,7,8,9)

Response: We thank the reviewer's constructive suggestion. Several papers showed that calcipotriol as the vitamin-D receptor (VDR) ligand efficiently deactivated CAFs which expressed VDR by reduced markers of fibrosis and the proliferation of CAFs (*Cell* 2014, 159: 80; *Gut* 2017, 66: 1449; *Journal of Steroid Biochemistry and Molecular Biology* 2013, 133: 12, et al). Accordingly, stroma-rich H22 tumor-bearing mice were intravenously injected with PBS, ICG, Cal/ICG, ICG@MPs, Cal@MPs or Cal/ICG@MPs twice every two days, followed with or without 808 nm laser irradiation. Consistently, besides the reduced expression of α -SMA, fibronectin and collagen-I, and deposition of collagen fibers, flow cytometric analysis showed that Cal@MPs and Cal/ICG@MPs significantly decreased the numbers of CAFs, namely podoplanin (PDPN)⁺CD140 α ⁺CD45⁻EpCAM⁻CD31⁻ cells or PDPN⁺ α -SMA⁺CD45⁻EpCAM⁻CD31⁻ cells, in tumor tissues compared with PBS group. Meanwhile, Cal@MPs- and Cal/ICG@MPs significantly inhibited the numbers of Ki67⁺ proliferative CAFs. However, Cal@MPs and Cal/ICG@MPs did not significantly induce apoptosis of myofibroblasts. These results showed that the Cal/ICG@MPs-induced decrease in CAF number might be due to the reduced proliferation, confirming that Cal/ICG@MPs efficiently deactivated, but not depleted CAFs. In addition, 808 nm laser irradiation did not markedly promote Cal/ICG@MPs-induced decrease in the numbers of CAFs and fibrosis markers, suggesting that Cal/ICG@MPs-triggered PTT did not further deplete CAFs. We added these data in Figure 4c, Supplementary Figure 18, 19 in the revised manuscript. Since normal fibroblasts also expressed VDR, Cal was reported to

inhibit the proliferation of normal fibroblasts (*Gut* 2017, 66: 1449; *Journal of Steroid Biochemistry and Molecular Biology* 2013, 133: 12, et al). In our work, we also found that the numbers of PDPN⁺α-SMA⁺CD45⁻EpCAM⁻CD31⁻ cells decreased in Cal@MPs- and Cal/ICG@MPs-treated group with or without 808 nm laser irradiation (Figure 1).

Figure 1. Numbers of PDPN⁺α-SMA⁺CD45⁻EpCAM⁻CD31⁻ cells in tumor tissues of stroma-rich H22 tumor-bearing mice after intravenous injection of PBS, ICG, Cal/ICG, ICG@MPs, Cal@MPs or Cal/ICG@MPs derived from H22 cells at the ICG dosage of 8 mg kg⁻¹ and Cal dosage of 120 μg kg⁻¹ twice every two days, followed with or without 808 nm laser irradiation (1.5 W cm⁻², 10 min) at 2 h after the last injection. Data are presented as means ± s.d. (n = 6, one-way ANOVA followed by Tukey's HSD post-hoc test).

The Cal/ICG@MPs with or without 808 nm laser irradiation-induced decrease in collagen content in tumor tissues of stroma-rich H22 tumor-bearing mice was shown in the Figure 4d and Supplementary Figure 21. According to the reviewer's suggestion, we supplemented the determination of the collagen content in tumor tissues of bilateral 4T1 orthotopic tumor models (Figures 8) and 4T1 metastasis tumor models (Figure 9) by Masson's trichrome staining. Consistently, Cal/ICG@MPs significantly decreased the deposition of collagen fibers in primary tumors of bilateral 4T1 orthotopic tumor-bearing mice and orthotopic tumors of metastatic 4T1 tumor-bearing mice (both treated tumor sides) with and without 808 nm laser irradiation, confirming that Cal/ICG@MPs efficiently remodeled tumor ECM. We added these data in Supplementary Figure 43 and 46 b,c in the revised manuscript.

e. In Fig 4I, it is not very evident that Cal/ICG MPs are significantly better than ICG MPs. Quantification of all replicates should be included to demonstrate this point convincingly.

Response: We thank the reviewer's constructive suggestion. Accordingly, to make the difference display more evidently, we changed the analysis of the ICG

fluorescence intensity at different depth of stroma-rich H22 tumors at 48 h after treatment with ICG, Cal/ICG, ICG@MPs or Cal/ICG@MPs. Consistent with the previous data, more Cal/ICG@MPs were significantly spread inside the tumor tissues compared with other groups. Furthermore, we used liver cancer patient-derived tumors and then treated with ICG, Cal/ICG, ICG@MPs or Cal/ICG@MPs for 48 h. Consistently, more ICG was observed in deep tumor tissues in Cal/ICG@MPs-treated groups compared with other groups, confirming that Cal/ICG@MPs-induced tumor ECM remodeling might contribute to their enhanced tumor penetration.

We added the new data in the Figure 4i, j and Supplementary Figure 25 in the revised manuscript.

f. The authors also show similar immune changes in the spleen and lymph nodes compared to tumours with better efficacy of Cal/ICG MPs compared to other groups. Why is the presence of Cal in the MPs be more effective than ICG alone? The authors should provide additional immunochemical and histological evidence if they believe it is also associated to CAFs/matrix, which should not be present at these sites.

Response: We thank the reviewer's critical question. In this work, stroma-rich H22 tumor-bearing mice were intravenously injected with PBS, ICG, Cal/ICG, ICG@MPs, Cal@MPs or Cal/ICG@MPs, followed with or without 808 nm laser irradiation. Due to the efficient tumor targeting of Cal/ICG@MPs and Cal-triggered CAF deactivation to remodel tumor ECM to enhance ICG tumor accumulation and penetration, Cal/ICG@MPs exhibited stronger photothermal effects against tumor cells upon 808 nm laser irradiation, resulting in more tumor antigen release to generate stronger antitumor immunity. The PTT-induced systemic antitumor immunity was reported in several papers, including *Nature Communications* 2021, 12: 742; *ACS Nano* 2019, 13:11967; *Advanced Materials* 2014, 26: 8154, et al. We think that the Cal/ICG@MPs-improved tumor immune microenvironment in the tumor draining lymph nodes and spleens upon 808 nm laser irradiation was due to the generation of strong systemic anticancer immunity.

g. Figure 6f should be quantified. From the images, it appears that α SMA is reduce also by ICG MPs.

Response: We thank the reviewer's constructive suggestion. Accordingly, we quantified the α -SMA expression in tumor tissues of stroma-rich H22 tumor-bearing mice after intravenous injection of PBS, ICG, Cal/ICG, ICG@MPs, Cal@MPs or Cal/ICG@MPs twice every two days, followed with 808 nm laser

irradiation. Consistently, Cal/ICG@MPs, but not ICG@MPs significantly decreased α -SMA expression.

We added the data in the Supplementary Figure 35 in the revised manuscript.

(2) Can the authors provide any data that this is relevant to the human disease? Clearly the immune aspect of the manuscript cannot be validated in vivo and would be challenging to address in vitro. The authors could, however, repeat some of the in vitro and/or in vivo experiments with at least one human cancer cell line (and fibroblast line) to at least show increased accumulation in the presence of Cal/ICG MPs compared to other groups etc.

Response: We thank the reviewer's constructive suggestion. Accordingly, we collected ICG@MPs, Cal@MPs and Cal/ICG@MPs derived from human hepatocellular carcinoma HepG2 cells. HepG2 cells were first treated with ICG, Cal/ICG, ICG@MPs or Cal/ICG@MPs, and the ICG fluorescence was determined by flow cytometry. Consistently, ICG@MPs and Cal/ICG@MPs exhibited higher ICG intracellular internalization than ICG or Cal/ICG, confirming that tumor cell-derived MPs enhanced cellular uptake of ICG. Furthermore, HepG2 cells were treated with PBS, ICG, Cal/ICG, ICG@MPs, Cal@MPs or Cal/ICG@MPs in the presence or absence of 808 nm laser irradiation. Stronger cytotoxicity was detected in the ICG@MPs- and Cal/ICG@MPs-treated HepG2 cells upon laser irradiation compared with free ICG- or Cal/ICG-treated group, confirming that Cal/ICG@MPs-triggered PTT induced strong cytotoxicity against HepG2 cells. Correspondingly, the percentage of CRT-positive cells in ICG@MPs- and Cal/ICG@MPs-treated groups upon 808 nm laser irradiation was significantly higher than free ICG- and Cal/ICG-treated groups, confirming that Cal/ICG@MPs-triggered PTT can induce strong ICD effects of HepG2 cells. Meanwhile, TGF- β -activated primary human lung fibroblasts were incubated with PBS, ICG, Cal, Cal/ICG, ICG@MPs, Cal@MPs or Cal/ICG@MPs and then fibronectin expression was determined by immunofluorescence analysis. Consistently, Cal/ICG@MPs and Cal@MPs exhibited stronger inhibition in the expression of fibronectin than free Cal and Cal/ICG, confirming that Cal/ICG@MPs efficiently deactivated CAFs in human CAFs.

Furthermore, liver cancer patient-derived tumors were treated with PBS, Cal/ICG, ICG@MPs, Cal@MPs or Cal/ICG@MPs for 48 h. Consistently, Cal@MPs and Cal/ICG@MPs significantly decreased the expression of α -SMA, fibronectin and collagen-I by immunofluorescence analysis, and reduced the deposition of collagen fibers by Masson's trichrome staining, suggesting that Cal/ICG@MPs efficiently remodel tumor ECM through CAF deactivation in

human tumor tissues. In addition, liver cancer patient-derived tumors were treated with ICG, Cal/ICG, ICG@MPs or Cal/ICG@MPs for 48 h and then determined ICG distribution in tumor tissues. Consistently, the strongest ICG fluorescence was detected in Cal/ICG@MPs-treated group. Meanwhile, more Cal/ICG@MPs were spread inside the tumor tissues compared with other groups, suggesting that Cal/ICG@MPs-induced tumor ECM remodeling might contribute to their enhanced tumor accumulation and penetration in human tumor tissues.

We added these data in Supplementary Figure 6, 15, 16 and 25 in the revised manuscript.

(3) The majority of flow data in respect to immune cell populations show specific markers within that population. Can the authors also include the total number (i.e. from live singlets) of the main populations central to their story (i.e. T cells, CD8⁺ T cells, Tregs, DCs)? This is particularly relevant for T cells and CD8⁺ T cells if the authors want to claim increased T cell infiltration, not only activation, in tumours and in vitro. Especially, since it appears that Cal/ICG MPs is better than ICG MPs in this regard, but not compared to other groups (e.g. Fig j-k).

Response: We thank the reviewer's constructive suggestion. Accordingly, we provided the ratios and numbers of T cells, DCs, Tregs and PMN-MDSCs, et al in tumor tissues of stroma-rich H22 tumor-bearing mice and 4T1 tumor-bearing mice after treatment. Consistent with the ratio change, Cal/ICG@MPs significantly increased the numbers of CD8⁺, activated CD8⁺ T cells and DCs, while decreased Tregs, PMN-MDSC and M2-like TAMs.

We added these data in Figure 6b-e, 8g-n and Supplementary Figure 31, 44 in the revised manuscript.

Minor points

(1) Could the authors please provide the data described on lines 123-126? I believe it is not included in the manuscript, although mentioned there.

Response: We thank the reviewer's constructive suggestion. Accordingly, we provided the optimization of preparation of Cal/ICG@MPs and added the drug loading efficiency of Cal/ICG@MPs in the Supplementary Table 1 and 2 in the revised manuscript.

(2) Can the authors please clarify the rationale in the methods for the different time point chosen for the experiments in vitro (e.g. 24h for DCs, 5 days DCs + T cells, ...).

Response: We thank the reviewer's critical question. In this work, we performed some experiments according to the references. For example, we measured DC maturation by treating BMDCs with the tumor cell supernatants for 24 h according to *Nature Communications* 2020, 11: 1985; *Journal of Clinical Investigation* 2018, 128: 644-654; *Nature Methods* 2018, 15: 183-186, et al. The matured DC were incubated with T cells for 5 days to stimulate T cell activation according to *Molecular Cancer* 2015, 14: 174, et al. We added the corresponding references in the Methods section in the revised manuscript.

(3) Can the authors include in their analysis of T cells also exhaustion markers to evaluate whether they decrease in parallel to the increase in activation markers.

Response: We thank the reviewer's constructive suggestion. Accordingly, we determined the ratios of CD8⁺PD-1⁺ T cells in stroma-rich H22 tumor-bearing mice after intravenous injection of PBS, ICG, Cal/ICG, ICG@MPs, Cal@MPs or Cal/ICG@MPs twice every two days, followed with or without 808 nm laser irradiation at tumor sites for 10 min. The results showed that ICG@MPs and Cal/ICG@MPs also significantly increased the ratios of CD8⁺PD-1⁺ T cells upon 808 nm laser irradiation, which might be due to the PTT-generated tumor antigen stimulation (*Nature Reviews Immunology* 2015, 15: 486). The expression of PD-1 might provide the possibility of combination treatment of Cal/ICG@MPs upon NIR laser irradiation with PD-1 antibody to further improve the anticancer effects in the future work.

We added these data in Supplementary Figure 31e in the revised manuscript.

(4) The authors should not utilise the term "CAFs" when talking about fibroblasts that have not been exposed to tumour cues, and instead should use the term "myofibroblasts".

Response: We thank the reviewer's constructive suggestion. Accordingly, we replaced CAFs with myofibroblasts when talking about fibroblasts that have not been exposed to tumour cues in the revised manuscript.

(5) In addition to CD45 and CD31, an epithelial marker (e.g. EpCAM and/or ECAD) should have also been included in the analysis shown in 4C as PDPN and PDGFRA can in some cases be expressed by cancer cells.

Response: We thank the reviewer's constructive suggestion. Accordingly, we determined the number of CAFs (PDPN⁺CD140 α ⁺CD45⁻EpCAM⁻CD31⁻ cells

and PDPN⁺α-SMA⁺CD45⁻EpCAM⁻CD31⁻ cells) in tumor tissues of stroma-rich H22 tumor-bearing mice after intravenous injection of PBS, Cal/ICG, ICG@MPs, Cal@MPs or Cal/ICG@MPs twice every two days, confirming that Cal@MPs and Cal/ICG@MPs significantly decreased the ratio of CAFs in tumor tissues.

We added these data in the Figure 4c and Supplementary Figure 18 in the revised manuscript.

(6) Could the authors please clarify whether Fig.6i is significant between Cal/ICG MPs and other groups?

Response: We thank the reviewer's constructive suggestion. Accordingly, we supplemented the statistical analysis of Figure 6i and it showed that there was significant difference between Cal/ICG@MPs with 808 nm laser irradiation-treated group compared with other groups at deep tumor tissues. We replaced Figure 6i with statistical analysis in the revised manuscript.

(7) Can the authors please provide zoomed in pictures for the tumours in Supp Fig 20 and clarify whether there is increased necrosis upon treatment with Cal/ICG MPs? The authors should clarify what they mean saying that the treatment is effective based on those images.

Response: We thank the reviewer's constructive suggestion. Accordingly, we zoomed the H&E staining images of tumors, which showed that the significant nucleus dissociation was detected in Cal/ICG@MPs-treated group upon 808 nm laser irradiation. To more clearly clarify that the Cal/ICG@MPs with 808 nm laser irradiation-treated group exhibited the strongest anticancer activity, TUNEL analysis of tumor tissues after treatment was performed using immunohistochemistry. Consistently, the maximum proportion of TUNEL-positive apoptotic cells in tumor tissues was detected in Cal/ICG@MPs-treated group upon 808 nm laser irradiation, further confirming that Cal/ICG MPs exhibited the excellent anticancer activity upon NIR laser irradiation.

We added these data in Supplementary Figure 36a, b in the revised manuscript.

(8) In the chemotaxis experiment (supp fig 14), can the authors also provide the flow analysis for the activation markers of the CD8⁺ T cells?

Response: We thank the reviewer's constructive suggestion. Accordingly, we determined the ratio of CD69⁺ and IFN-γ⁺ (the activation markers) in CD8⁺ T cells in the chemotaxis experiment in which the cell supernatants from CAFs treated with PBS, ICG, Cal, Cal/ICG, ICG@MPs, Cal@MPs or Cal/ICG@MPs were added into the bottom chambers, and CD3⁺ T cells were seeded in the top chambers. The results showed that Cal/ICG@MPs-treated CAFs did not markedly affect the ratios of CD69⁺ and IFN-γ⁺ in CD8⁺ T cells compared with

other groups, suggesting that Cal/ICG@MPs-induced CAF deactivation contributes to CD8⁺ T cell recruitment, but not CD8⁺ T cell activation.

We added these data in the Supplementary Figure 29b, c in the revised manuscript.

(9) The change in solid stress of tumours was only shown for the 4T1 models, can the authors provide this for the H22 models or state in the main text that this has not been assessed in that context?

Response: We thank the reviewer's constructive question. Accordingly, the solid stress of tumors was determined in stroma-rich H22 tumor-bearing mice after intravenous injection of PBS, Cal/ICG, ICG@MPs, Cal@MPs or Cal/ICG@MPs twice every two days. Consistently, treatment with Cal@MPs and Cal/ICG@MPs significantly decreased the solid stress of tumor tissues of H22 tumor models.

We added the data in Supplementary Figure 23a in the revised manuscript.

(10) Do Cal/ICG MPs affect cancer cell and fibroblast' proliferation in vivo more than other groups?

Response: We thank the reviewer's constructive suggestion. Accordingly, stroma-rich H22 tumor-bearing mice were intravenously injected with PBS, Cal/ICG, ICG@MPs, Cal@MPs or Cal/ICG@MPs twice every two days, followed with or without 808 nm laser irradiation at the tumor sites for 10 min. Consistently, Cal/ICG@MPs significantly inhibited the proliferation of CAFs with or without 808 nm laser irradiation, as evidenced by the decreased ratio of Ki67⁺ cells in PDPN⁺CD140α⁺CD45⁻EpCAM⁻CD31⁻ cells, demonstrating that Cal/ICG@MPs efficiently inhibited the proliferation of CAFs. Meanwhile, flow cytometric analysis also showed that Cal@MPs and Cal/ICG@MPs significantly decreased the ratio of Ki67⁺ cells in tumor cells (CD45⁻EpCAM⁻ cells), might be due to the Cal/ICG@MPs-induced CAF deactivation. 808 nm laser irradiation further promoted the Cal/ICG@MPs-induced decrease in the ratio of Ki67⁺ cells in tumor cells Cal/ICG@MPs, suggesting that Cal/ICG@MPs efficiently inhibited the proliferation of tumor cells upon 808 nm laser irradiation.

We added these data in the Supplementary Figure 18b and 36c in the revised manuscript.

Reviewers' Comments:

Reviewer #1:

Remarks to the Author:

The authors addressed my concerns and made the changes. It can be accepted in the current form.

Reviewer #2:

Remarks to the Author:

The authors have successfully addressed my concerns and have provided data on human tumors, further supporting the potential importance of their concept for stroma remodeling.

Reviewer #3:

Remarks to the Author:

Dear editor

The manuscript can be accepted at present.

Reviewer #4:

Remarks to the Author:

The authors have largely addressed my concerns. I still, however, believe that "deactivation" is not the most appropriate term in this context and should be rephrased, as analysis of quiescent fibroblast markers (e.g. lipid droplets in HSCs) is missing. While the treatment may not be actively "depleting" already present fibroblasts (since no apoptosis is detected), the reduction in fibroblast number and fibroblast proliferation indicates a CAF-poor microenvironment following treatment. I would advise to use "targeting" instead of "deactivation" in the title and elsewhere, and to discuss these 2 possibilities in the manuscript.

As a minor note, since I believe the authors are not including the data presented in Figure 1 of the rebuttal (although it could be useful), PDPN+aSMA- fibroblasts are not necessarily normal fibroblasts, but could be low-myofibroblastic, inflammatory CAFs, which have been described in various cancer types.

Finally, the authors should double check that they have included all significant stats and p values (e.g. is there a difference between ICG@MPs+Laser and Cal/ICG@MPs+Laser in Supp Fig 37c?).

Response to Reviewer 4

(1) The authors have largely addressed my concerns. I still, however, believe that “deactivation” is not the most appropriate term in this context and should be rephrased, as analysis of quiescent fibroblast markers (e.g. lipid droplets in HSCs) is missing. While the treatment may not be actively “depleting” already present fibroblasts (since no apoptosis is detected), the reduction in fibroblast number and fibroblast proliferation indicates a CAF-poor microenvironment following treatment. I would advise to use “targeting” instead of “deactivation” in the title and elsewhere, and to discuss these 2 possibilities in the manuscript.

Response: We thank the reviewer’s constructive suggestion. Accordingly, we changed the title of the manuscript. Since our results showed that more Cal/ICG@MPs were internalized into tumor cells compared with CAFs, to avoid ambiguity we did not use “targeting”, and changed “deactivating” to “regulating” in the title and elsewhere in the revised manuscript. Meanwhile, according to the reviewer’s suggestion, we added these information in the Discussion section in the revised manuscript.

(2) As a minor note, since I believe the authors are not including the data presented in Figure 1 of the rebuttal (although it could be useful), PDPN⁺αSMA⁻ fibroblasts are not necessarily normal fibroblasts, but could be low-myofibroblastic, inflammatory CAFs, which have been described in various cancer types.

Response: We thank the reviewer’s constructive suggestion. Accordingly, we added the data on the numbers of PDPN⁺α-SMA⁻CD45⁻EpCAM⁻CD31⁻ cells after treatment with Cal/ICG@MPs in the presence or absence of 808 nm laser irradiation in the Supplementary Figure 18c in the revised manuscript.

(3) Finally, the authors should double check that they have included all significant stats and p values (e.g. is there a difference between ICG@MPs+Laser and Cal/ICG@MPs+Laser in Supp Fig 37c?).

Response: We thank the reviewer’s constructive suggestion. Accordingly, we have checked the whole statistics and p values in the revised manuscript.